# Hyper-LLaVA: Hyperbolic Uncertainty-aware Modality-Balanced Routing for Multimodal Continual Instruction Tuning

Kunlun Xu [* 1]   Yanqin Zhang [* 1]   Wenwen Qiang [2 3]   Jiahuan Zhou [† 1]

## Abstract

Multimodal Continual Instruction Tuning (MCIT) aims to exploit the incrementally accumulated knowledge to process multimodal inputs of diverse tasks, where parameter routing plays an important role. State-of-the-art methods rely on sample-to-task center similarity and cross-modal fusion with equal weight during routing. However, such solutions face two fundamental flaws: (1) Within each modality, the sample-to-task center distance is sub-optimal for routing since the abundant intra-task diversity information is underleveraged. (2) Different modalities exhibit varying reliability across tasks, where the modality with inter-task ambiguity can easily misguide the routing result. To address these problems, we propose Hyperbolic Uncertainty-aware Modality-Balanced Routing (Hyper-LLaVA) to improve parameter routing capacity based on cross-modality task feature uncertainty modeling. Specifically, to improve intra-modality task matching, Hyper-LLaVA accesses the sample to task distribution similarity in the Hyperbolic space. Besides, to alleviate the degradation brought by unreliable modality, Hyper-LLaVA quantifies the task matching ambiguity within each modality to achieve adaptive balancing between task matching across modalities. Based on the complementary intra- and inter-modality task matching enhancement, our Hyper-LLaVA outperforms state-of-the-art approaches by large margins. Our source code is available at https://github.com/zhoujiahuan1991/ICML2026-Hyper-LLaVA

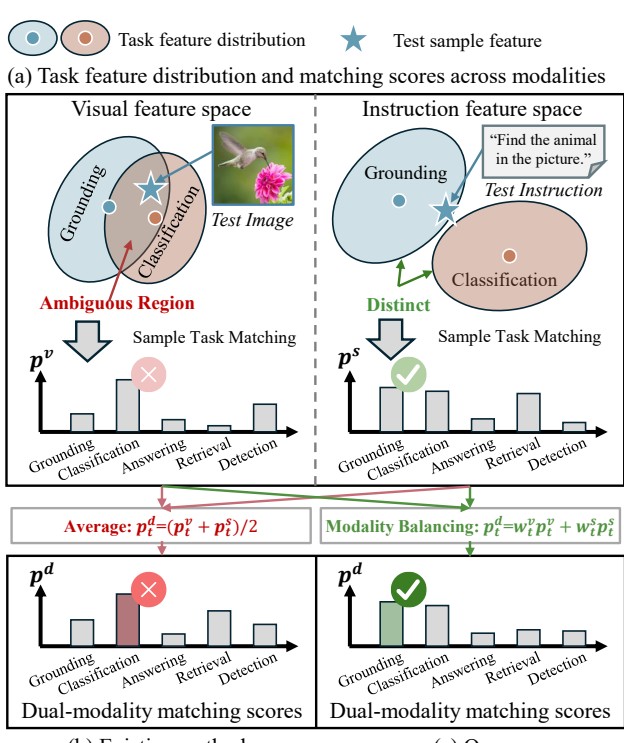

*Figure 1.* (a) Within different modalities, the task feature regions may be ambiguous or distinct. The ambiguous task feature regions can often result in high erroneous task matching scores. and distinct task feature distribution may also generate low correct task matching scores. (b) Existing methods are vulnerable to erroneous task matching scores due to simply averaging the dual-modality predictions. (c) Our method aims to seek an adaptive balance between modalities to generate reliable task selection.

## 1. Introduction

Multi-modal Large Language Models (MLLMs) (Yin et al., 2024) extend the reasoning capabilities of Large Language Models (LLMs) (Touvron et al., 2023) to the visual domain by integrating visual encoders (Radford et al., 2021). To bridge the gap between large-scale pre-training and downstream task, instruction tuning (Zhao et al., 2025; Ouyang et al., 2022) has been introduced, where MLLMs are fine-tuned on paired (*instruction*, *image*, *response*) data to enhance their ability to follow user commands. However, in practical scenarios, it is often infeasible to collect data for

---

[*]Equal contribution  [1]Wangxuan Institute of Computer Technology, Peking University, Beijing, China [2]Institute of Software Chinese Academy of Sciences, Beijing, China [3]University of Chinese Academy of Sciences, Beijing, China. Correspondence to: Jiahuan Zhou <jiahuanzhou@pku.edu.cn>.

*Proceedings of the 43rd International Conference on Machine Learning*, Seoul, South Korea. PMLR 306, 2026. Copyright 2026 by the author(s).

all downstream tasks in advance. Instead, task data typically arrive sequentially, requiring the model to continuously acquire new capabilities over time. This setting gives rise to Multimodal Continual Instruction Tuning (MCIT) (He et al., 2023; Zheng et al., 2024), which aims to enable MLLMs to progressively learn new tasks while preserving performance on previously learned ones.

Since MLLMs have already acquired abundant fundamental knowledge during large-scale pre-training, recent MCIT methods are typically built upon Mixture-of-Experts architectures (Wang et al., 2025; Guo et al., 2025). Usually, the pretrained parameters are kept frozen, while task-specific experts, *e.g.*, prompts (Zeng et al., 2025; Li et al., 2024b; Xu et al., 2025; Zhang et al., 2025a; Xu et al., 2026b) or Low-Rank Adaptation (LoRA) modules (Chen et al., 2024), are introduced to encode incrementally arriving task knowledge. Such methods perform intra-modality task matching by measuring the distance between a test sample and task centers, and then generate the final task matching results by averaging the predictions from the visual and instruction (textual) modalities.

However, such paradigms suffer from two fundamental limitations. First, within each modality, routing based solely on the distance between a sample and a task center is suboptimal, as the abundant intra-task diversity information is largely underexploited (Xu et al., 2024; Zhou et al., 2025). Second, as illustrated in Figure 1 (a), the same task may exhibit varying levels of reliability across different modalities due to task distribution overlap. In task-ambiguous regions, test samples may produce erroneously high task-matching scores, whereas samples located near distinct task distribution boundaries may yield low scores despite being correctly matched. When these inter-modality scores are naively averaged, unreliable predictions from one modality can dominate the final decision, thereby misleading task prediction and degrading overall performance.

To address these problems, we propose a novel MCIT framework named Hyperbolic Uncertainty-aware Modality-Balanced Routing (Hyper-LLaVA), which improves both intra-modality matching and inter-modality aggregation from an uncertainty-aware perspective. Given the training data of each incoming task, Hyper-LLaVA not only learns the corresponding task-specific expert but also estimates the statistical properties of the task data distribution. During inference, to enhance intra-modality task prediction, both sample features and task distributions are projected into a hyperbolic space, where similarity is measured using the uncertainty quantified by the Poincaré distance, which provides improved robustness to complex distributional variations. Furthermore, to mitigate erroneous task predictions caused by unreliable modalities, Hyper-LLaVA estimates inter-task separation to quantify task ambiguity within each modal-

ity. Based on this estimation, modality-wise weights are adaptively assigned to each task through a dual-modality ambiguity balancing mechanism, as shown in Figure 1(b). By jointly enhancing intra-modality similarity estimation and inter-modality score aggregation, Hyper-LLaVA achieves significantly more accurate parameter routing, outperforming state-of-the-art MCIT approaches by a large margin.

To sum up, our contributions include:
(1) We propose Hyper-LLaVA, a novel MCIT framework that addresses two key limitations of existing methods: poor modeling of intra-task distributional diversity, and unreliable inter-modality aggregation under task ambiguity.
(2) Our approach jointly enhances routing by embedding samples and task distributions into hyperbolic space for robust intra-modality similarity measurement, and adaptively balancing modalities based on ambiguity estimated from inter-task separation.
(3) Extensive experiments on two challenging MCIT benchmarks show Hyper-LLaVA outperforms state-of-the-art MCIT methods by a large margin.

## Conflict of Interest Disclosure

The authors declare that they have no known competing financial interests or personal relationships that could have appeared to influence the work reported in this paper.

## 2. Related Work

### 2.1. Multimodal Large Language Models

The rapid advancement of Large Language Models (LLMs) (Touvron et al., 2023) has paved the way for Multimodal Large Language Models (MLLMs), which extend the reasoning capabilities of LLMs to the visual domain. Proprietary models such as GPT-4V (Achiam et al., 2023) and Gemini (Team et al., 2023)have demonstrated exceptional performance in understanding and reasoning about complex visual inputs. In the open-source community, MLLMs typically adopt a modular architecture comprising a vision encoder (e.g., CLIP (Radford et al., 2021; Liu et al., 2025b; Xu et al., 2026a) or SigLIP (Zhai et al., 2023)), a pretrained LLM backbone, and a vision-language connector. Early prominent works like BLIP-2 (Li et al., 2023), and InstructBLIP (Dai et al., 2023) utilized a Q-Former to bridge the modality gap, effectively compressing visual information into query tokens for the LLM. MiniGPT-4 (Zhu et al., 2023) demonstrated that aligning a frozen visual encoder with a frozen LLM using a single projection layer could yield strong emergent abilities. Building on this streamlined design, LLaVA (Liu et al., 2023) and its enhanced version LLaVA-1.5 (Liu et al., 2024) employed a Multi-Layer Perceptron (MLP) connector and leveraged GPT-4 generated data for visual instruction tuning, establishing a robust

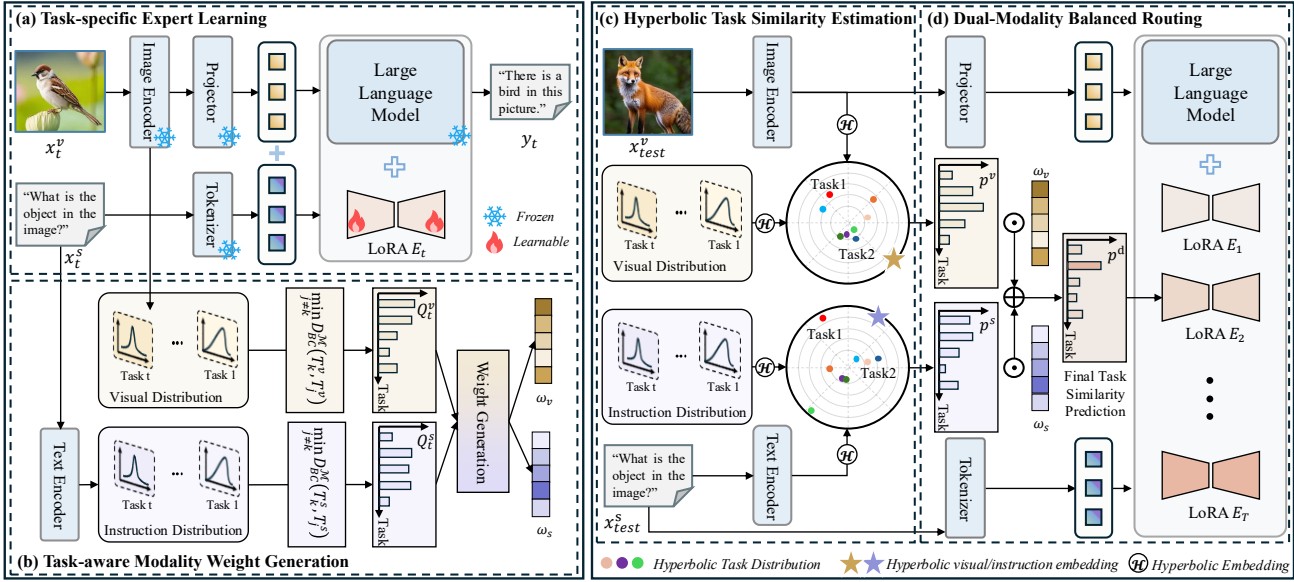

*Figure 2.* **The overview of our proposed Hyper-LLaVA framework.** The process consists of two stages: **(Left) The Training Stage**, comprising **(a)** standard instruction tuning with task-specific LoRA experts and **(b)** a Task-aware Modality Reliability Estimation module. This module quantifies task separability via Bhattacharyya Distance ($D_{BC}$) to derive adaptive modality weights ($\omega_v, \omega_t$). **(Right) The Inference Stage**, specifically **(c)** the Hyperbolic Uncertainty-aware Modality-Balanced Routing. We map features into the Poincaré ball to capture task uncertainty and fuse modality similarities ($S_v, S_t$) using the learned weights for precise expert selection.

baseline for general-purpose multimodal tasks.

## 2.2. Continual Instruction Tuning

While instruction tuning significantly enhances the zero-shot capabilities of MLLMs, maintaining these abilities across a continuous stream of changing tasks remains a formidable challenge due to the phenomenon of catastrophic forgetting. Consequently, Multimodal Continual Instruction Tuning (MCIT) has emerged as a critical research frontier, aiming to balance the stability of previously acquired knowledge with the plasticity required for new tasks. To rigorously evaluate this capability, comprehensive benchmarks such as CoIN (Chen et al., 2024) and CITB (Zhang et al., 2023b) have been established, alongside domain-specific settings like VQACL (Zhang et al., 2023a), revealing that standard fine-tuning strategies suffer severe performance degradation in sequential settings. To mitigate this, a dominant trend involves parameter-efficient architecture expansion, particularly using Mixture-of-Experts frameworks. Approaches like MoELoRA (Chen et al., 2024), Continual-LLaVA (Cao et al., 2024), and LLaVA-c (Liu et al., 2025a) utilize dynamic adapters, while more advanced frameworks like CL-MoE (Huai et al., 2025) and SMoLoRA (Wang et al., 2025) introduce dual-momentum updates or dual-forgetting mechanisms to refine expert routing. Other structural innovations include BranchLoRA (Zhang et al., 2025b), LoRA in LoRA (Che et al., 2025), and Progressive LoRA (Yu et al., 2025),

which hierarchically expand the parameter space to accommodate new tasks. Additionally, HiDe-LLaVA (Guo et al., 2025) proposes a hierarchical decoupling framework, performing task-specific expansion on top layers while fusing task-general knowledge in lower layers. In parallel, regularization and optimization strategies have been proposed to constrain learning trajectories, such as GNSP (Peng et al., 2025), which projects gradients into the null space of previous tasks, and SEFE (Chen et al., 2025), which targets specific types of forgetting via tailored regularization. Furthermore, ModalPrompt (Zeng et al., 2025) exploits dual-modality guidance for task retrieval.

## 3. The Proposed Method

### 3.1. Preliminaries

**Multi-modal Continual Instruction Tuning (MCIT).** MCIT focuses on a sequential learning scenario where a Multi-modal Large Language Model (MLLM), parameterized by $\theta$, is trained on a stream of $T$ tasks $\mathcal{T} = \{T_1, T_2, \ldots, T_T\}$. Each task $T_t$ consists of a dataset $\mathcal{D}_t = \{(x_{t,i}^v, x_{t,i}^s, y_{t,i})_{i=1}^{N_t}\}$, where $x^v$ represents the visual input, $x^s$ is the textual instruction, and $y$ is the target response. The goal is to sequentially adapt the model to task $T_t$ while retaining the knowledge of previous tasks $T_{1:t-1}$, without accessing their training data.

## 3.2. Task-specific Expert Learning

Following previous works (Guo et al., 2025), we adopt Low-Rank Adaptation (LoRA) as a Parameter-Efficient Fine-Tuning (PEFT) strategy to adapt the Multimodal Large Language Model (MLLM) to downstream tasks while mitigating computational overhead. LoRA freezes the pre-trained model weights $W_0 \in \mathbb{R}^{d \times k}$ and injects trainable rank decomposition matrices into the linear layers. Specifically, for each adapted layer, LoRA introduces two low-rank matrices $B \in \mathbb{R}^{d \times r}$ and $A \in \mathbb{R}^{r \times k}$, with rank $r \ll \min(d, k)$. The modified forward pass is computed as:

$$W' = W_0 + \Delta W = W_0 + BA \qquad (1)$$

where $W_0$ remains frozen, $A$ is initialized from a Gaussian distribution, and $B$ is initialized to zero. To strictly isolate task-specific knowledge and prevent catastrophic forgetting, we maintain a pool of LoRA experts $\mathcal{E} = \{E_1, \ldots, E_T\}$. Each expert $E_t$ corresponds to the specific parameters learned exclusively for task $t$. Consequently, the weight update for a specific task $t$ is expressed as:

$$\Delta W_t = B_t A_t \qquad (2)$$

where $B_t \in \mathbb{R}^{d \times r}$ and $A_t \in \mathbb{R}^{r \times k}$ denote the task-specific matrices. Given an input image $x^v$ and instruction $x^s$, the model is optimized to generate response $y$ by minimizing the autoregressive cross-entropy loss, updating only the parameters of the current expert $E_t$. This isolation prevents catastrophic forgetting of previous tasks $T_{1:t-1}$ at the parameter level.

## 3.3. Task-aware Modality Weight Generation

While task-specific experts ensure parameter-level isolation, robustly routing inputs to the correct expert during inference is non-trivial due to the modality imbalance problem. Static fusion strategies often fail when one modality is noisy or non-discriminative. To address this, we propose an adaptive weighting mechanism governed by the Bhattacharyya Coefficient (BC), which is computed at the conclusion of each training stage.

We quantify the discriminative power of a specific modality by evaluating the separability of task data distributions within its feature space. Specifically, at the end of training stage $t$, we compute the pair-wise Bhattacharyya Distance $D_{BC}$ between every pair of learned tasks in the current pool. To facilitate tractable computation, we assume the feature distributions of each task follow a multivariate Gaussian distribution with independent dimensions. Under this assumption, $D_{BC}$ between task $T_k$ and $T_j$ can be analytically derived by summing the distances across all feature dimensions $D$:

$$D_{BC}(T_k, T_j) = -\ln(BC(T_k, T_j))$$
$$= -\sum_{d=1}^{D} \ln \left( BC_d \left( T_k^{(d)}, T_j^{(d)} \right) \right) \qquad (3)$$

where the single-dimension component $\ln(BC_d)$ is defined as:

$$\ln(BC_d) = -\frac{1}{4} \frac{(\mu_{k,d} - \mu_{j,d})^2}{\sigma_{k,d}^2 + \sigma_{j,d}^2}$$
$$+ \frac{1}{2} \ln \left( \frac{2\sigma_{k,d}\sigma_{j,d}}{\sigma_{k,d}^2 + \sigma_{j,d}^2} \right) \qquad (4)$$

where, $\mu_{k,d}$ and $\sigma_{k,d}^2$ denote the mean and variance of task $k$ at the $d$-th dimension. Based on these pair-wise distances, we derive the separation vectors $\mathbf{Q}_t^v$ and $\mathbf{Q}_t^s$ for visual and instruction modalities, respectively. The element corresponding to a given task $T_k$ represents its discriminability score $m_k$, defined as the distance to its nearest neighbor among other tasks:

$$m_k^{\mathcal{M}} = \min_{j \neq k} D_{BC}^{\mathcal{M}}(T_k, T_j), \quad \mathcal{M} \in \{v, s\} \qquad (5)$$

A larger $m_k$ implies that task $k$ occupies a highly unique region in the feature space of modality $\mathcal{M}$, suggesting that this modality is reliable for identifying task $k$. Due to magnitude inconsistencies, we perform normalization within each modality using the average separation scale:

$$\hat{m}_k^{\mathcal{M}} = \frac{m_k^{\mathcal{M}}}{\frac{1}{t} \sum_{j=1}^{t} m_j^{\mathcal{M}} + \epsilon}, \quad \mathcal{M} \in \{v, s\} \qquad (6)$$

Finally, the normalized scores are passed through a temperature-scaled Softmax function to yield the adaptive modality weights $(\omega_v, \omega_s)$. The calculation method for the $k$-th task is as follows:

$$\omega_k^v, \omega_k^s = \text{Softmax}(\hat{m}_k^v, \hat{m}_k^s) \qquad (7)$$

## 3.4. Hyperbolic Task Similarity Estimation

Standard Euclidean metrics for similarity estimation often suffer from the "gravity well" effect, where high-variance general tasks overshadow low-variance specific tasks. To enable precise expert selection, we propose embedding task distributions in a Hyperbolic space, specifically the Poincaré ball model $(\mathbb{D}^d, g_p)$. For each learned task $T_k$, we construct a embedding function $\mathcal{H}(\boldsymbol{\mu}, V)$ to project its Euclidean statistics (centroid $\boldsymbol{\mu}_k$ and aggregate variance $V_k = \sum \boldsymbol{\sigma}_k^2$) into the Poincaré ball:

$$z_k = \mathcal{H}(\boldsymbol{\mu}_k, V_k) = r_k \cdot \frac{\boldsymbol{\mu}_k}{\|\boldsymbol{\mu}_k\|_2} \qquad (8)$$

*Table 1.* Comparison of the proposed Hyper-LLaVA method with existing approaches in CoIN Benchmark.

| Method | Venue | Accuracy on Each Task (%) | | | | | | | | Aggregate Results (%) | | | |
|---|---|---|---|---|---|---|---|---|---|---|---|---|---|
| | | SQA | TVQA | ImgNet | GQA | VizWiz | Gron | VQAv2 | OVQA | MFN↑ | MAA↑ | MFT↑ | BWT↑ |
| Zero-shot | - | 49.91 | 2.88 | 0.33 | 2.08 | 0.90 | 0.00 | 0.68 | 0.17 | 7.12 | - | - | - |
| Multi-task | - | 56.77 | 49.35 | 95.55 | 56.65 | 53.90 | 30.09 | 59.50 | 55.65 | 57.18 | - | - | - |
| LwF | *TPAMI 2017* | 74.45 | 49.70 | 39.30 | 52.00 | 50.45 | 7.05 | 62.25 | 47.80 | 47.88 | 59.71 | 68.13 | -20.26 |
| EWC | *NAS 2017* | 75.20 | 55.36 | 67.50 | 54.70 | 52.90 | 15.40 | 64.45 | 35.05 | 52.57 | 61.63 | 65.59 | -13.02 |
| O-LoRA | *CVPR 2023* | 76.90 | 42.65 | 15.85 | 40.25 | 45.10 | 0.30 | 54.35 | 54.00 | 41.18 | 56.28 | 56.99 | -15.82 |
| MoE-LoRA | *NeurIPS 2024* | 76.90 | 42.65 | 15.85 | 40.25 | 45.10 | 0.30 | 54.35 | 54.00 | 41.18 | 56.28 | 56.99 | -15.82 |
| HiDe-LLaVA | *ACL 2025* | 73.20 | 56.92 | 69.28 | 61.33 | 50.76 | 59.18 | 67.12 | 64.76 | 62.82 | - | - | - |
| ModalPrompt | *EMNLP 2025* | 68.42 | 56.40 | 41.13 | 61.11 | 50.13 | 36.69 | 66.90 | 59.68 | 55.06 | 59.24 | 56.39 | -1.33 |
| SEFE | *ICML 2025* | 75.35 | 58.66 | 83.10 | 54.25 | 48.85 | 16.75 | 65.35 | 66.25 | 58.57 | 63.04 | 69.02 | -10.45 |
| Hyper-LLaVA | *This Paper* | 77.10 | 59.02 | 93.49 | 60.54 | 57.88 | 51.81 | 66.76 | 65.13 | **66.47** | **68.79** | 68.77 | -2.30 |

The key point lies in encoding the task's uncertainty $V_k$ into the hyperbolic radius $r_k \in [0, 1)$. We employ an adaptive scaling strategy to ensure robustness across varying feature scales during incremental learning:

$$r_k = \exp\left(-\frac{\gamma}{\bar{V}_t + \epsilon} \cdot V_k\right) \quad (9)$$

where $\bar{V}_t$ is the moving average of variances across all known tasks, and $\gamma$ is a scaling constant. This formulation ensures that specific tasks are mapped to the boundary ($r_k \to 1$), while general tasks are mapped near the center ($r_k \to 0$). For an incoming test input $x$, we treat it as a deterministic point distribution with zero variance, embedding it to $z = \mathcal{H}(x, 0)$ with a radius $r \approx 1$. The similarity between the test input and an expert $E_k$ is defined by the negative Poincaré distance:

$$D_p(x, E_k) = -\text{arccosh}\left(1 + 2\frac{\|z - z_k\|^2}{(1 - \|z\|^2)(1 - \|z_k\|^2)}\right) \quad (10)$$

For a specific task located at the boundary ($r_k \to 1$), the denominator $(1 - \|z_k\|^2)$ in the distance formula approaches zero. This causes the distance to grow exponentially with even minor angular deviations, strictly enforcing specificity. Conversely, general tasks near the center remain tolerant to variations. Proof as shown in **Theorem A.1**. This geometric property effectively resolves the "gravity well" issue, preventing high-variance tasks from overshadowing specific ones.

### 3.5. Dual-Modality Balanced Routing

In the final inference phase, we integrate the hyperbolic similarity measures with the task-aware modality weights to perform robust expert routing. For a test input $x = (x_{test}^v, x_{test}^s)$, we independently compute the Poincaré distance for all available experts in both visual ($D_p^v$) and instruction ($D_p^s$) modalities. These scores are then fused via

our Adaptive Balancing mechanism. To ensure dynamic range alignment between modalities, we utilize adaptive temperatures ($\tau_v, \tau_s$) derived from the standard deviation of the scores:

$$p^v = \text{Softmax}(D_p^v/\tau_v), \quad p^s = \text{Softmax}(D_p^s/\tau_s) \quad (11)$$

The final dual modal decision score $p^d$ is computed as the weighted sum of the probability distributions from both modalities, using the pre-computed reliability weights ($\omega_v, \omega_s$):

$$p^d = \omega_v \odot p^v + \omega_s \odot p^s \quad (12)$$

where $\odot$ denotes element-wise multiplication. To rigorously justify this adaptive weighting strategy, we introduce the following theoretical bounds regarding estimation error.

**Lemma 3.1** (Optimality of Variance-Inverse Weighting). *Let $q^v$ and $q^s$ be the observed similarity scores from visual and instruction modalities, modeled as the true signal perturbed by independent Gaussian noise with variances $\sigma_v^2$ and $\sigma_s^2$, respectively. The estimation error of a linear fusion strategy is minimized when the weights are inversely proportional to the modality uncertainty: $w^* = \sigma_s^2/(\sigma_v^2 + \sigma_s^2)$.*

Building on this lemma, we establish the theoretical basis for our task-aware modality weighting:

**Theorem 3.2** (Superiority of BC-based Fusion). *Under the assumption of approximately isotropic Gaussian feature distributions, the Bhattacharyya Distance $D_{BC}$ scales inversely with feature variance ($D_{BC} \propto 1/\sigma^2$). Therefore, the Dual-Modality Balanced Routing strategy, which assigns weights based on $D_{BC}$, effectively approximates the optimal inverse-variance fusion derived in **Lemma 3.1**.*

The proof of Lemma 3.1 and Theorem 3.2 are provided in Appendix A. Finally, based on the optimized decision score $p^d$, the expert $E_{k^*}$ with the highest similarity is selected to perform the specific task reasoning and response generation.

# 4. Experiments

## 4.1. Experimental Setup

**Datasets**. We use the CoIN (Chen et al., 2024) and UCIT (Guo et al., 2025) as our benchmark. CoIN contains eight vision-language tasks: VQAv2 (Goyal et al., 2017), ScienceQA (SQA) (Lu et al., 2022), TextVQA (TVQA) (Singh et al., 2019), ImageNet (ImgNet) (Deng et al., 2009), VizWiz (Gurari et al., 2018),GQA (Hudson & Manning, 2019), REC-COCO (Kazemzadeh et al., 2014; Mao et al., 2016), OCR-VQA (OVQA) (Mishra et al., 2019). UCIT (Guo et al., 2025) is designed to address the potential data leakage issue in CoIN (Chen et al., 2024), where downstream tasks might overlap with the MLLM's pre-training data. UCIT contains six vision-language tasks, including ArxivQA (Li et al., 2024a), CLEVR-Math (Lindström & Abraham, 2022), IconQA (Lu et al., 2021),ImageNet-R (Hendrycks et al., 2021), VizWiz-caption (Gurari et al., 2018).

**Evaluation Metrics**. we evaluate the two benchmarks with four metrics: (1)Mean Fine-tune Accuracy (MFT), the average accuracy of each task immediately after it is learned, which represents the upper-bound performance or plasticity of the model. (2)Mean Final Accuracy (MFN), the average accuracy of all tasks after the complete training sequence, which reflects the final knowledge retained. (3)Mean Average Accuracy (MAA), the average performance of all learned tasks at every incremental step. This measures the stability of the model throughout the entire life cycle. (4) Backward Transfer (BWT), the average drop in performance for previous tasks after learning new ones, measuring catastrophic forgetting. The definition of these metrics are provided in our Appendix C.

**Compared Methods**. We compare Hyper-LLaVA with a comprehensive set of baselines that cover both traditional continual learning and recent multimodal continual instruction tuning methods. The traditional continual learning baselines include LwF (Li & Hoiem, 2017), which applies knowledge distillation, and EWC (Kirkpatrick et al., 2017), which constrains important parameters. We also include O-LoRA (Wang et al., 2023), which learns tasks in orthogonal subspaces. In addition, several recent multimodal continual instruction tuning approaches are included for comparison. MoELoRA (Chen et al., 2024) employs a mixture-of-experts adapter design. ModalPrompt (Zeng et al., 2025) exploits dual-modality prompts for task adaptation. HiDe-LLaVA (Guo et al., 2025) adopts a hierarchical decoupling strategy, while SEFE (Chen et al., 2025) focuses on style diversification and regularization. Furthermore, Zero-shot performance (inference without fine-tuning) and Joint Training performance (multi-task learning on all data) are reported as the lower and upper bounds, respectively.

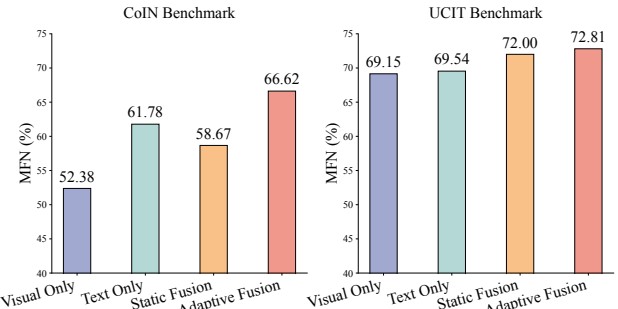

*Figure 3.* Ablation study on modality fusion strategies.

**Implementation details**. Consistent with the settings in CoIN (Chen et al., 2024) , we use LLaVA-v1.5-7B (Liu et al., 2024) as our backbone, which integrates a Vicuna-7B LLM with a CLIP-L/14-336 (Radford et al., 2021) visual encoder. LoRA (Hu et al., 2022) is applied to all linear layers of the LLM, with a rank of $r = 64$, $\alpha = 128$ in CoIN benchmark and $r = 48$, $\alpha = 96$ in UCIT benchmark, following the previous works (Guo et al., 2025). During training, the cosine decay schedule is adopted and the learning rates is set to $2e - 4$ for LoRA modules. We train for 1 epoch per task with a batch size of 24, and the warm-up ratio to 0.03. The scaling coefficient $\gamma$ is set to 1.0. All experiments are conducted on 8 NVIDIA H800 GPUs.

## 4.2. Comparison with Existing Approaches

**Results on CoIN.** Table 1 reports the performance comparison on the CoIN benchmark. Classical continual learning methods, such as LwF and EWC, show limited final performance and suffer from severe catastrophic forgetting due to continual parameter updates. When compared with the state-of-the-art MoE-based method HiDe-LLaVA, our approach achieves a **3.65%** improvement in MFN. This gain is attributed to the proposed Hyperbolic Task Similarity Estimation mechanism and Dual-Modality Balancing scheme, which together enhance expert routing reliability. Besides, compared with SEFE, which relies on LoRA merging, Hyper-LLaVA yields **7.90%** and **5.75%** improvements in MFN and MAA, respectively. These improvements are attributed to the robust expert routing strategy of Hyper-LLaVA, which avoids cross-expert fusion that may otherwise introduce knowledge interference.

**Results on UCIT.** Table 2 summarizes the results on the UCIT benchmark. Compared with state-of-the-art MoE-based frameworks (O-LoRA, MoE-LoRA, HiDe-LLaVA, and ModalPrompt), Hyper-LLaVA achieves at least **8.46%** and **3.85%** improvements in MFN and MAA, respectively. These gains demonstrate the effectiveness of the proposed Hyperbolic Uncertainty-aware Modality-Balanced Routing mechanism in enhancing task prediction reliability. When

*Table 2.* Comparison of the proposed Hyper-LLaVA method with existing approaches in UCIT Benchmark.

| Method | Venue | Accuracy on Each Task (%) | | | | | | Aggregate Results (%) | | | |
|---|---|---|---|---|---|---|---|---|---|---|---|
| | | ImgNetR | ArxivQA | VizWiz | IconQA | CLEVR | Flickr | MFN↑ | MAA↑ | MFT↑ | BWT↑ |
| Zero-shot | - | 16.27 | 53.73 | 38.39 | 19.20 | 20.63 | 41.88 | 31.68 | - | - | - |
| Multi-task | - | 91.67 | 90.83 | 57.87 | 78.43 | 76.63 | 61.72 | 76.19 | - | - | - |
| LwF | *TPAMI 2017* | 40.27 | 75.93 | 42.76 | 44.38 | 37.43 | 56.34 | 49.52 | - | - | - |
| EWC | *NAS 2017* | 39.05 | 77.88 | 43.24 | 45.33 | 39.72 | 55.94 | 50.20 | - | - | - |
| O-LoRA | *CVPR 2023* | 77.50 | 78.07 | 44.50 | 63.13 | 64.73 | 58.16 | 64.35 | 78.02 | 76.01 | -13.99 |
| MoE-LoRA | *NeurIPS 2024* | 70.07 | 77.70 | 44.69 | 50.03 | 54.03 | 57.34 | 58.98 | 75.08 | 71.17 | -14.63 |
| HiDe-LLaVA | *ACL 2025* | 84.03 | 90.73 | 44.43 | 58.93 | 41.37 | 54.25 | 62.29 | 77.32 | 69.96 | -9.20 |
| ModalPrompt | *EMNLP 2025* | 74.43 | 92.00 | 55.92 | 44.27 | 53.97 | 43.67 | 60.70 | 70.57 | 60.76 | -0.05 |
| SEFE | *ICML 2025* | 80.83 | 78.00 | 47.01 | 69.63 | 65.83 | 57.92 | 66.54 | 78.76 | 75.98 | -11.33 |
| Hyper-LLaVA | *This Paper* | 87.20 | 93.70 | 57.24 | 75.20 | 65.60 | 57.92 | **72.81** | **81.87** | 78.03 | -5.22 |

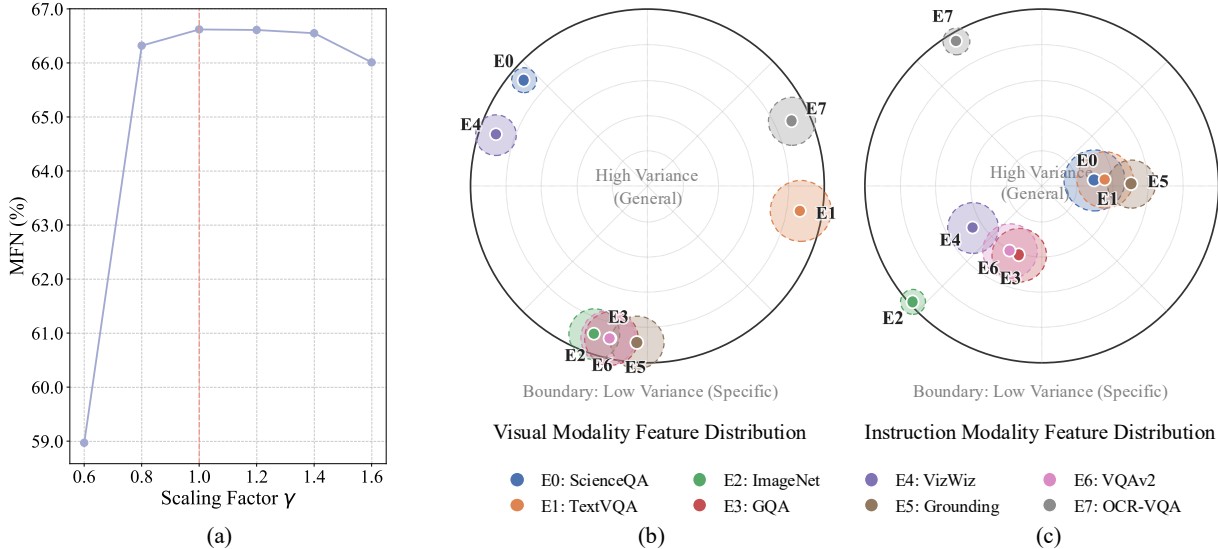

*Figure 4.* Ablation study on modality fusion strategies (a), Task feature distributions in the Poincaré ball (b) and (c).

*Table 3.* Ablation study on the effectiveness of Hyperbolic Metric and Adaptive Fusion (metric: MFN).

| Metric Space | Fusion Strategy | MFN (%) |
|---|---|---|
| Euclidean | Static | 56.75 |
| Poincaré | Static | 62.67 (+5.92) |
| Euclidean | Adaptive | 66.42 (+9.67) |
| **Poincaré** | **Adaptive (Ours)** | **66.62 (+9.87)** |

compared with the expert-merging-based method SEFE, improvements of **6.27 %** in MFN and **3.11 %** in MAA are achieved, further verifying the superiority of the proposed strategy in expert selection. Moreover, Hyper-LLaVA establishes state-of-the-art MFT performance on the UCIT benchmark, with at least a **2.02 %** improvement over existing methods, which further supports its effectiveness in

selecting appropriate experts.

It is noted that inferior BWT performance, which reflects forgetting, is observed compared with ModalPrompt. This behavior arises because ModalPrompt introduces additional constraints to explicitly mitigate forgetting on previous tasks. While such constraints improve stability, they restrict model plasticity and may limit adaptation to new data. In contrast, the overall performance metrics MFN and MAA indicate that Hyper-LLaVA achieves a more favorable stability–plasticity trade-off than ModalPrompt.

### 4.3. Ablation Studies

**Impact of Core Components.** To verify the impact of core Components on model performance improvement, we isolate the contributions of the hyperbolic metric and BC-based adaptive fusion in Table 3. Simply swapping the Euclidean

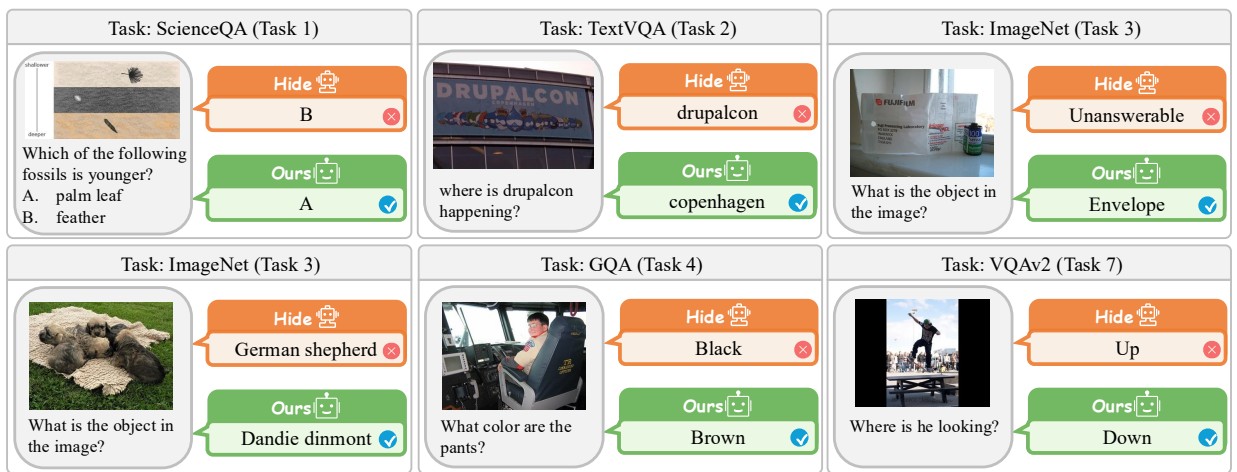

*Figure 5.* Visualization of model predictions across different tasks in comparison with state-of-the-art Hide (Guo et al., 2025).

metric for Poincaré distance yields a consistent performance bump (e.g., **+5.92%** with Static Fusion), confirming that hyperbolic geometry provides a superior embedding for capturing task hierarchy and uncertainty. However, the true leap in performance comes from BC-based adaptive fusion. This delivers a massive gain of **+9.67%**, which signals that the imbalance of modality is not a minor inconvenience but a primary bottleneck in MCIT. The best results are achieved only when both components are combined, validating their complementary nature.

**Impact of Modality Fusion Strategies.** Figure 3 offers a fascinating glimpse into how different fusion strategies behave within the Poincaré space. On the CoIN benchmark, the system actually performs better using only instructions than it does with static fusion. This results in a clear case of modality pollution: blindly averaging visual signals introduces noise that degrades decision-making. This result flips on UCIT, where static fusion beats the unimodal baselines, indicating that vision and instruction are cooperating effectively. Crucially, BC-based adaptive fusion reigns supreme in both scenarios on CoIN and UCIT. This proves that the proposed BC-based weighting acts as an effective gatekeeper, suppressing harmful modalities when they introduce noise and amplifying cross-modal synergy when it offers a competitive advantage.

**Hyperparameter Sensitivity Analysis.** In order to ensure robustness across varying feature scales during continual learning, we introduce the adaptive scaling strategy, where the selection of the scaling coefficient plays an essential role in the model performance. In Figure 4 (a), we analyze the impact of different scaling coefficients $\gamma$. Overall, performance follows a bell-shaped curve. The small scaling coefficients lead to insufficient task separation, whereas large coefficients reduce hierarchical distinctions. This indicates that the critical point $\gamma = 1.0$ provides an effective

balance to take advantage of hyperbolic geometry.

### 4.4. Visualization

**Task Geometry in the Poincaré Ball.** Figures 4 (b) and (c) show the visualization of task distribution embeddings in the Poincaré disk, which exposes striking cross-modal asymmetries. Similar tasks clump together in the visual domain, while the instruction domain gives a different result. ScienceQA, for instance, is distinct in visual space but buried deep within a central cluster in instruction space, indicating a clear sign of low instruction discriminability. Conversely, due to the low variance of its own data distribution, ImageNet hugs the boundary of the instruction space, geometrically encoding its high specificity. These structural disparities validate our dual-modality balanced fusion strategy: the model must dynamically pivot to the modality where the task is most separable.

**Visualization of model predictions.** To intuitively demonstrate the efficacy of Hyper-LLaVA in mitigating task interference, we present qualitative comparisons in Figure 5. The result of ImageNet illustrates the preservation of fine-grained discriminative capability. The baselines hallucinate generic categories, while Hyper-LLaVA retains the capacity to identify a Dandie Dinmont. We also see the "gravity well" effect in action, baselines frequently misroute specific queries to high-variance experts, resulting in generic Unanswerable outputs, whereas Hyper-LLaVA boundary mapping enforces the necessary specificity. Beyond classification, complex reasoning scenarios further underscore the robustness of the framework. By calibrating the signal strength of vision versus text on the fly, the model successfully navigates diverse reasoning paths without succumbing to the noise of a dominant but irrelevant modality.

# 5. Conclusion

In this paper, we presented Hyper-LLaVA, a novel framework for Multimodal Continual Instruction Tuning (MCIT) that addresses the fundamental limitations of existing parameter routing mechanisms from an uncertainty-aware perspective. Specifically, to improve intra-modality task matching, Hyper-LLaVA accesses the sample to task distribution similarity in the Hyperbolic space. Besides, to alleviate the degradation brought by unreliable modality, Hyper-LLaVA quantifies the task matching ambiguity within each modality to achieve adaptive balancing between task matching across modalities. Based on the complementary intra- and inter-modality task matching enhancement, Hyper-LLaVA achieves significantly more accurate parameter routing, outperforming state-of-the-art MCIT approaches by a large margin. Our results underscore the importance of modeling task uncertainty and leveraging non-Euclidean geometries for routing in MLLMs. Future work could explore the extension of this hyperbolic uncertainty modeling to a broader range of multimodal foundation models and experts.

# Acknowledgements

This work was a research achievement of National Engineering Research Center of New Electronic Publishing Technologies, and was supported by the National Natural Science Foundation of China (62376011, 62506355), and the National Key R&D Program of China (2024YFA1410000).

# Impact Statement

This paper presents work whose goal is to advance the field of Machine Learning. There are many potential societal consequences of our work, none which we feel must be specifically highlighted here.

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

# A. Theorem

## A.1. Proof of Optimality for Modality-Adaptive Fusion

**Lemma 3.1** (Optimality of Variance-Inverse Weighting). *Let $q^v$ and $q^s$ be the observed similarity scores from visual and instruction modalities, modeled as the true signal perturbed by independent Gaussian noise with variances $\sigma_v^2$ and $\sigma_s^2$, respectively. The estimation error of a linear fusion strategy is minimized when the weights are inversely proportional to the modality uncertainty: $w^* = \sigma_s^2/(\sigma_v^2 + \sigma_s^2)$.*

*Proof.* We define a linear fusion strategy as $\hat{q} = wq^v + (1-w)q^s$, where $w \in [0, 1]$ is the weight assigned to the visual modality. The mean squared error (MSE) of this estimator is equivalent to its variance:

$$\mathcal{L}(w) = \mathrm{Var}(\hat{q}) = w^2\sigma_v^2 + (1-w)^2\sigma_s^2 \tag{13}$$

For the Static Fusion strategy, the weights are fixed at $w = 0.5$. Substituting this into Eq (13):

$$\mathcal{L}_{static} = 0.5^2\sigma_v^2 + 0.5^2\sigma_s^2 = \frac{\sigma_v^2 + \sigma_s^2}{4} \tag{14}$$

For the Adaptive Fusion strategy, we seek the optimal weight $w^*$ that minimizes the error $\mathcal{L}(w)$. Differentiating Eq (13) with respect to $w$ and setting it to zero:

$$\frac{\partial \mathcal{L}(w)}{\partial w} = 2w\sigma_v^2 - 2(1-w)\sigma_s^2 = 0 \tag{15}$$

Solving for $w$, we obtain the optimal weight which is inversely proportional to the variance:

$$w^* = \frac{\sigma_s^2}{\sigma_v^2 + \sigma_s^2} \tag{16}$$

Substituting $w^*$ back into Eq. (13), the minimum error for the adaptive strategy is:

$$\mathcal{L}_{adaptive} = \left(\frac{\sigma_s^2}{\sigma_v^2 + \sigma_s^2}\right)^2\sigma_v^2 + \left(\frac{\sigma_v^2}{\sigma_v^2 + \sigma_s^2}\right)^2\sigma_s^2 = \frac{\sigma_v^2\sigma_s^2}{\sigma_v^2 + \sigma_s^2} \tag{17}$$

To compare the two strategies, we examine the difference $\Delta = \mathcal{L}_{static} - \mathcal{L}_{adaptive}$:

$$\Delta = \frac{\sigma_v^2 + \sigma_s^2}{4} - \frac{\sigma_v^2\sigma_s^2}{\sigma_v^2 + \sigma_s^2} = \frac{(\sigma_v^2 + \sigma_s^2)^2 - 4\sigma_v^2\sigma_s^2}{4(\sigma_v^2 + \sigma_s^2)} \tag{18}$$

Expanding the numerator:

$$(\sigma_v^2 + \sigma_s^2)^2 - 4\sigma_v^2\sigma_s^2 = \sigma_v^4 + 2\sigma_v^2\sigma_s^2 + \sigma_s^4 - 4\sigma_v^2\sigma_s^2 = (\sigma_v^2 - \sigma_s^2)^2 \tag{19}$$

Since $(\sigma_v^2 - \sigma_s^2)^2 \geq 0$ and the variances are positive, it follows that $\Delta \geq 0$. Therefore:

$$\mathcal{L}_{adaptive} \leq \mathcal{L}_{static} \tag{20}$$

The equality holds only when $\sigma_v^2 = \sigma_s^2$. This completes the proof of the lemma. $\square$

**Theorem 3.2** (Superiority of BC-based Fusion). *Under the assumption of approximately isotropic Gaussian feature distributions, the Bhattacharyya Distance $D_{BC}$ scales inversely with feature variance ($D_{BC} \propto 1/\sigma^2$). Therefore, the Dual-Modality Balanced Routing strategy, which assigns weights based on $D_{BC}$, effectively approximates the optimal inverse-variance fusion derived in **Lemma 3.1**.*

*Proof.* While Lemma 3.1 establishes that the optimal fusion weight is inversely proportional to the noise variance $\sigma^2$ (as shown in Eq. (16)), estimating this variance directly during inference is intractable. We prove that the Bhattacharyya Distance $D_{BC}$ serves as a reliable proxy for the inverse variance.

Consider two task feature distributions in a given modality, modeled as multivariate Gaussians $P_i(x) = \mathcal{N}(\boldsymbol{\mu}_i, \boldsymbol{\Sigma}_i)$ for $i \in \{1, 2\}$. The Bhattacharyya distance between $P_1$ and $P_2$ admits the closed-form expression:

$$D_{BC}(P_1, P_2) = \frac{1}{8}(\boldsymbol{\mu}_1 - \boldsymbol{\mu}_2)^\top \boldsymbol{\Sigma}^{-1}(\boldsymbol{\mu}_1 - \boldsymbol{\mu}_2) + \frac{1}{2} \ln \frac{\det \boldsymbol{\Sigma}}{\sqrt{\det \boldsymbol{\Sigma}_1 \det \boldsymbol{\Sigma}_2}}, \tag{21}$$

where $\boldsymbol{\Sigma} = \frac{1}{2}(\boldsymbol{\Sigma}_1 + \boldsymbol{\Sigma}_2)$.

Under the common assumption that task features are approximately isotropic with equal variance, i.e., $\boldsymbol{\Sigma}_1 = \boldsymbol{\Sigma}_2 = \sigma^2 \mathbf{I}$, the second (log-determinant) term in Eq (21) vanishes. The equation simplifies to:

$$D_{BC}(P_1, P_2) = \frac{1}{8\sigma^2} \|\boldsymbol{\mu}_1 - \boldsymbol{\mu}_2\|_2^2. \tag{22}$$

Equation (22) demonstrates that, for a fixed inter-task separation $\|\boldsymbol{\mu}_1 - \boldsymbol{\mu}_2\|_2$, the Bhattacharyya distance scales inversely with the variance:

$$D_{BC} \propto \frac{1}{\sigma^2} \tag{23}$$

Consequently, a larger $D_{BC}$ corresponds to lower feature uncertainty (smaller $\sigma^2$) and higher discriminability. By assigning weights based on $D_{BC}$ via our DIW module, we are effectively approximating the optimal weighting scheme $w^* \propto 1/\sigma^2$ derived in Lemma 3.1. Therefore, our BC-based fusion strategy yields a lower estimation error than static fusion. □

## A.2. Proof of the Boundary Sensitivity in Poincaré Ball.

**Theorem A.1** (Gradient Explosion at the Boundary). *Let $d_{\mathbb{D}}(\mathbf{u}, \mathbf{v})$ be the Poincaré distance between two points $\mathbf{u}, \mathbf{v} \in \mathbb{D}^n$. Let $\mathbf{u}$ be fixed and $\mathbf{v} = \mathbf{u} + \boldsymbol{\delta}$ be a small perturbation. As the norm $\|\mathbf{u}\| \to 1$ (approaching the boundary), the rate of change of the hyperbolic distance with respect to the Euclidean perturbation $\boldsymbol{\delta}$ approaches infinity. Conversely, as $\|\mathbf{u}\| \to 0$ (approaching the center), the rate of change approaches the Euclidean rate.*

*Proof.* The Riemannian metric tensor $g_{\mathbf{x}}$ of the Poincaré ball model at a point $\mathbf{x}$ is given by a conformal scaling of the Euclidean metric $g^E$:

$$g_{\mathbf{x}} = \lambda_{\mathbf{x}}^2 g^E, \quad \text{where } \lambda_{\mathbf{x}} = \frac{2}{1 - \|\mathbf{x}\|^2} \tag{24}$$

The infinitesimal hyperbolic distance $ds_{\mathbb{D}}$ corresponds to the Euclidean distance $ds_E$ scaled by the conformal factor $\lambda_{\mathbf{x}}$:

$$ds_{\mathbb{D}} = \lambda_{\mathbf{x}} \|d\mathbf{x}\|_2 = \frac{2\|d\mathbf{x}\|_2}{1 - \|\mathbf{x}\|^2} \tag{25}$$

Now, consider the sensitivity, defined as the ratio of hyperbolic change to Euclidean change:

$$\text{Sensitivity}(\mathbf{x}) = \frac{ds_{\mathbb{D}}}{ds_E} = \frac{2}{1 - \|\mathbf{x}\|^2} \tag{26}$$

**Case 1: General Tasks (Center).** For tasks mapped near the center, $\|\mathbf{x}\| \to 0$. The sensitivity becomes:

$$\lim_{\|\mathbf{x}\| \to 0} \text{Sensitivity}(\mathbf{x}) = 2 \tag{27}$$

This implies that near the center, the hyperbolic distance behaves similarly to Euclidean distance (up to a constant factor of 2). The metric space is "flat" and tolerant to perturbations, suitable for high-variance general tasks.

**Case 2: Specific Tasks (Boundary).** For tasks mapped near the boundary, $\|\mathbf{x}\| \to 1$. The sensitivity becomes:

$$\lim_{\|\mathbf{x}\| \to 1} \text{Sensitivity}(\mathbf{x}) = \infty \tag{28}$$

This implies that at the boundary, even an infinitesimal Euclidean displacement $\|d\mathbf{x}\|_2$ results in an infinitely large hyperbolic distance. This geometric property enforces the high selectivity required for specific, low-variance tasks, effectively preventing them from being overshadowed by general tasks. □

### A.3. Connection between Bhattacharyya Coefficient and Classification Error.

**Theorem A.2** (Bhattacharyya Bound on Bayes Error). *Let $P_1(x)$ and $P_2(x)$ be the probability density functions of two tasks in a specific modality. The discriminability of these two tasks is bounded by the Bhattacharyya Coefficient $BC(P_1, P_2)$. Specifically, the probability of misclassifying a sample from Task 1 as Task 2 (Bayes Error rate $P_e$) is upper-bounded by the Bhattacharyya Coefficient.*

*Proof.* The Bayes error rate $P_e$ for a binary classification problem between two distributions with equal prior probabilities is defined as the integral of the minimum of the two densities:

$$P_e = \int \min(P_1(x), P_2(x)) dx \tag{29}$$

Using the inequality $\min(a, b) \leq \sqrt{ab}$ for any non-negative real numbers $a, b$, we can write:

$$\min(P_1(x), P_2(x)) \leq \sqrt{P_1(x)P_2(x)} \tag{30}$$

Integrating both sides over the domain $\mathcal{X}$:

$$\int \min(P_1(x), P_2(x)) dx \leq \int \sqrt{P_1(x)P_2(x)} dx \tag{31}$$

The right-hand side is exactly the definition of the Bhattacharyya Coefficient $BC(P_1, P_2)$. Thus, we arrive at the Bhattacharyya bound:

$$P_e \leq BC(P_1, P_2) \tag{32}$$

Since the Bhattacharyya Distance is defined as $D_{BC} = -\ln(BC)$, minimizing the overlap $BC$ (or maximizing the distance $D_{BC}$) directly minimizes the upper bound of the classification error. Therefore, by assigning higher weights to the modality with a larger $D_{BC}$ (smaller $BC$), our adaptive fusion strategy theoretically prioritizes the feature space with the lower potential for routing error. $\square$

# B. Algorithm

---

**Algorithm 1** Pipeline of Hyper-LLaVA

---

1: **Input:** Sequential task datasets $\mathcal{D}_t = \{(x_{t,i}^v, x_{t,i}^s, y_{t,i})_{i=1}^{N_t}\}$.
2: **Initialize:** Base MLLM $\Theta$, Empty Expert Pool $\mathcal{E} = \emptyset$.
3: **for** $t = 1$ **to** $T$ **do**
4:      # Training Stage: Acquire new knowledge
5:      Initialize LoRA expert $E_t$ for task $t$.
6:      **for** minibatch $\mathcal{B} \in \mathcal{D}_t$ **do**
7:          Update $E_t$ via autoregressive loss $\mathcal{L}_{CE}$.
8:      **end for**
9:      Add $E_t$ to Expert Pool $\mathcal{E}$.
10:      # Statistics Update
11:      Extract features for $\mathcal{D}_t$ using frozen encoders.
12:      Compute Euclidean centroids $\boldsymbol{\mu}_t$ and variances $V_t$.
13:      Update adaptive scaling factor $\gamma$ based on $\{V_1, \ldots, V_t\}$.
14:      # Hyperbolic Task Similarity Estimation
15:      Compute pair-wise Bhattacharyya Distances among $\{T_1, \ldots, T_t\}$.
16:      Derive separation vectors $Q$ and modality weights $\omega$ via Eq(7).
17:      # Inference Stage (for a test input $x$)
18:      **for** each test input $x = (x_{test}^v, x_{test}^s)$ **do**
19:          # Hyperbolic Embedding
20:          Map experts to Poincaré ball: $\mathbf{z}_k = \mathcal{H}(\boldsymbol{\mu}_k, V_k)$ $(r_k \propto V_k^{-1})$.
21:          Map query to Poincaré boundary: $\mathbf{z} = \mathcal{H}(x, 0)$ $(r \to 1)$.
22:          # Dual-Modality Balanced Routing
23:          **for** $k = 1$ **to** $t$ **do**
24:              Calculate Poincaré similarities $q^v, q^s$ via Eq(10).
25:              Calculate fuse scores $q^d$ via Eq (12)
26:          **end for**
27:          Select optimal expert: $k^* = \arg\max_k q^d$.
28:          Generate response $y_i$ using Expert $E_{k^*}$.
29:      **end for**
30: **end for**

---

## C. Evaluation Metrics

To comprehensively evaluate the performance of the model in the continual learning setting, we employ four standard metrics. Let $T$ denote the total number of tasks in the sequential stream. We define an accuracy matrix $A \in \mathbb{R}^{T \times T}$, where $A_{t,i}$ represents the test accuracy on task $i$ after the model has finished learning task $t$. Based on this formulation, the metrics are defined as follows:

**1. Mean Fine-tune Accuracy (MFT).** This metric evaluates the model's *plasticity*, measuring its capacity to acquire new knowledge from the current task. It is calculated as the average accuracy of each task $i$ immediately after it has been learned:

$$\text{MFT} = \frac{1}{T} \sum_{i=1}^{T} A_{i,i}. \tag{33}$$

A higher MFT indicates that the model effectively adapts to new domains without being overly constrained by previous knowledge.

**2. Mean Final Accuracy (MFN).** To assess the overall proficiency of the model upon the conclusion of the continual learning curriculum, we report the average accuracy across all $T$ tasks after the final training stage:

$$\text{MFN} = \frac{1}{T} \sum_{i=1}^{T} A_{T,i}. \tag{34}$$

This metric reflects the final knowledge retention and is the primary indicator of the model's ultimate performance.

**3. Mean Average Accuracy (MAA).** While MFN focuses solely on the final state, MAA provides a holistic view of the model's *stability* throughout the entire lifecycle. It is computed as the average of the performance at every incremental step $t$:

$$\text{MAA} = \frac{1}{T} \sum_{t=1}^{T} \left( \frac{1}{t} \sum_{i=1}^{t} A_{t,i} \right). \tag{35}$$

A high MAA suggests that the model maintains consistent performance across both new and old tasks during the intermediate stages of training, rather than fluctuating significantly.

**4. Backward Transfer (BWT).** This metric quantifies the influence of learning new tasks on the performance of previously acquired tasks. It is defined as the average difference between the final accuracy of a task and its initial accuracy immediately after learning:

$$\text{BWT} = \frac{1}{T} \sum_{i=1}^{T} (A_{T,i} - A_{i,i}). \tag{36}$$

A negative BWT value indicates *catastrophic forgetting*, where performance degrades over time. Conversely, a positive BWT implies positive backward transfer, where learning new tasks facilitates the performance on older tasks.

# D. Inter-task Relevance Analysis

Figure 6 visualizes inter-task relevance using the Log Bhattacharyya Coefficient. The visual modality exhibits clear block-diagonal structures, where tasks sharing image sources (e.g., GQA, Grounding, VQAv2) form dense clusters, implying high visual overlap and potential routing ambiguity. In contrast, the instruction modality shows a sparse structure with low off-diagonal relevance, indicating sharp task boundaries. This contrast motivates our dual-modality balanced routing strategy, which down-weights modalities with high task overlap and emphasizes more discriminative ones.

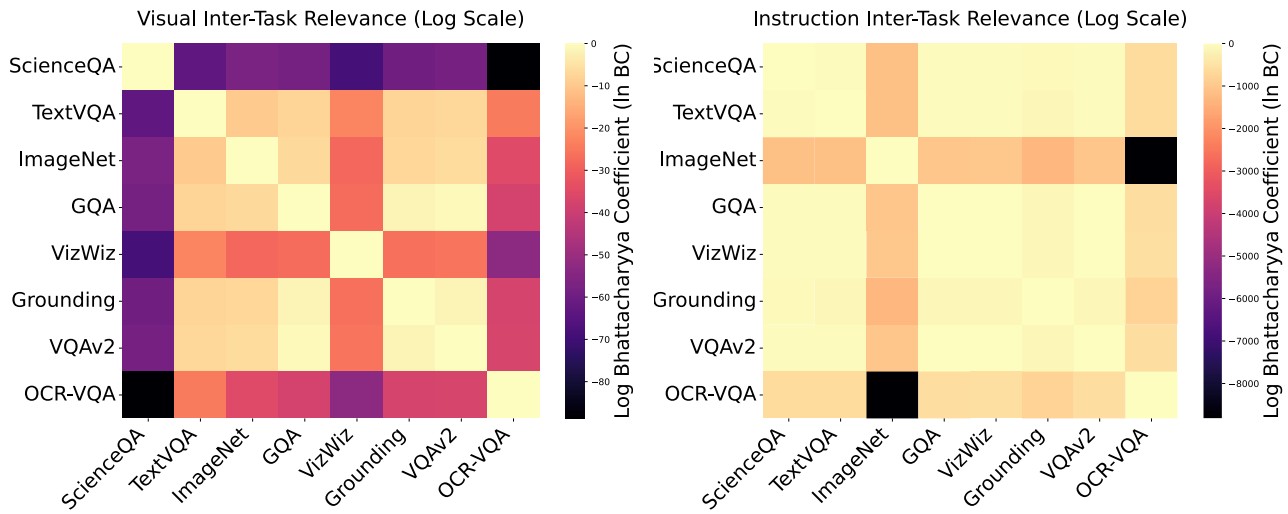

*Figure 6.* Log-scale Inter-task Relevance Matrices on CoIN Benchmark

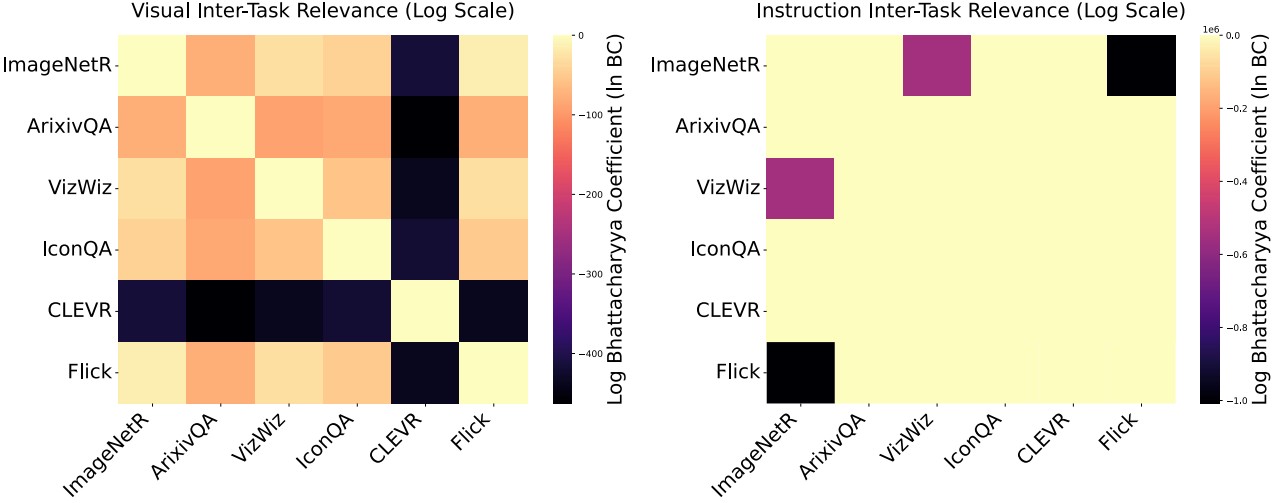

*Figure 7.* Log-scale Inter-task Relevance Matrices on UCIT Benchmark

# E. Task-aware Modality Weight Analysis

The Figure 8 illustrates the dynamic assignment of modality weights on CoIN (left) and UCIT (right) benchmarks. Row $t$ represents the weight distribution used for routing at the end of stage $t$. We observe significant variation in $\omega_v$ across tasks (columns), confirming that the discriminative power of the visual modality is highly task-dependent. For instance, in CoIN, ScienceQA maintains a high throughout all stages, indicating its strong visual distinctiveness, whereas ImageNet shows extremely low visual weights, correctly deferring to the instruction modality as intended by our hyperbolic boundary mapping. In addition, the values of weights are not all close to 0 or 1. Many task ID predictions require a comprehensive judgment based on the integration of two modalities. This adaptive mechanism effectively prevents modality dominance by "noisy" modalities.

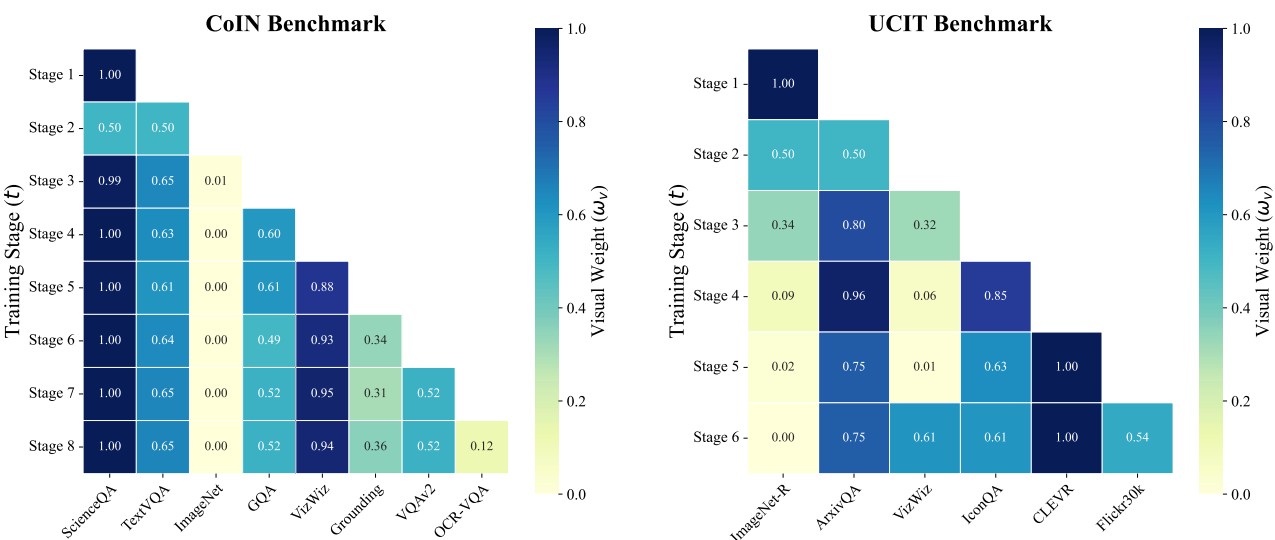

*Figure 8.* Evolution of Task-aware Modality Weight

## F. More Comparison Results

Figure 9 presents further comparisons with the state-of-the-art HiDe model on the challenging ScienceQA, ImageNet and TextVQA benchmarks, which encompass diverse social and natural vision–instruction understanding scenarios. The results indicate that more consistent and robust adaptation is achieved by our model.

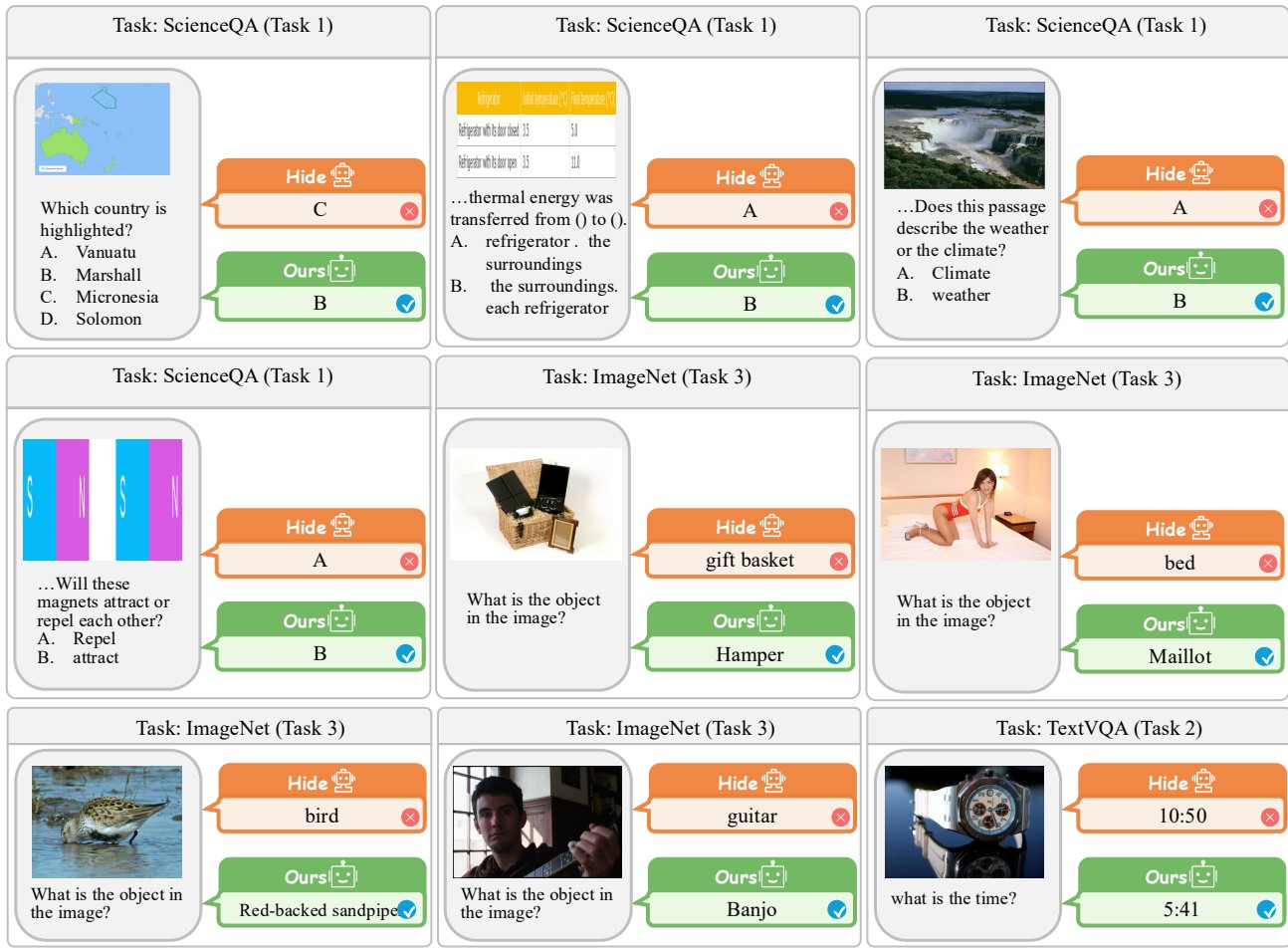

*Figure 9.* Visualization of model predictions across different tasks in comparison with state-of-the-art Hide (Guo et al., 2025).

