# OpenReview forum: "Hyper-LLaVA: Hyperbolic Uncertainty-aware Modality-Balanced Routing for Multimodal Continual Instruction Tuning"
_ICML.cc/2026/Conference — ICML 2026 regular_

### Official Review · Reviewer_tw6i · 2026-03-08

**Soundness:** 3
**Presentation:** 3
**Significance:** 3
**Originality:** 3
**Overall Recommendation:** 4
**Confidence:** 4

**Summary:**

This paper introduces Hyper-LLaVA, a novel routing framework for Multimodal Continual Instruction Tuning (MCIT). The authors propose that existing routing mechanisms suffer from poor intra-task diversity modeling and unreliable inter-modality aggregation. To address these, Hyper-LLaVA estimates task distribution statistics and maps both sample features and task prototypes into hyperbolic space (Poincaré ball) for similarity estimation, leveraging hyperbolic geometry to handle uncertainty. Furthermore, a Dual-Modality Balanced Routing mechanism is proposed to adaptively weight modalities based on inter-task separation (ambiguity). Experimental results on two MCIT benchmarks demonstrate that Hyper-LLaVA outperforms several SOTA baselines.

**Compliance With Llm Reviewing Policy:**

Affirmed.

**Final Justification:**

Because the authors' rebuttal have addressed all of my concerns, so I decide to raise my score.

**Key Questions For Authors:**

1.Almost all equations in the manuscript lack terminal punctuation marks, which is standard in formal academic typesetting.

2.In Table 1 and Table 2, the best and second-best results for the MFT and BWT metrics are not highlighted , making it difficult to parse the comparative results quickly.

3.On line 413, "figure 4" is written in lowercase, whereas all other figure citations in the text follow the capitalized "Figure" format. Please ensure consistency throughout the manuscript.

**Limitations:**

Please see Weaknesses.

**Strengths And Weaknesses:**

**Strengths**

1.Incorporating task distribution statistics into the routing process allows the model to explicitly account for intra-task diversity, which is often neglected in sample-only routing.

2.The use of hyperbolic geometry for similarity measurement is well-motivated, as it naturally captures hierarchical or complex structures and provides a robust metric for uncertainty-aware task identification.

3.The ambiguity-balanced mechanism, based on inter-task separation, effectively mitigates the risk of a single unreliable modality misleading the global routing decision.

4.The method achieves competitive performance across two benchmarks with comprehensive ablation studies supporting the effectiveness of the individual components.

**Weaknesses**

1.The authors identify two critical limitations of existing methods in their first contribution: (a) poor modeling of intra-task diversity and (b) unreliable inter-modality aggregation. However, the manuscript lacks targeted experimental evidence or visualization to directly prove that these are indeed the primary failure modes of current SOTA methods. Quantitative evidence for these "limitations" is required to justify the complexity of the proposed solution.

2.For every new task, Hyper-LLaVA introduces a new set of LoRA modules without any mechanism for module reuse or distillation. As the task sequence grows, the linear increase in parameters raises significant concerns regarding memory consumption and storage scalability. The lack of sub-linear parameter growth designs limits the method's long-term utility.

3.The experiments are restricted to the LLaVA-v1.5-7B model. The authors are encouraged to validate their method on more modern and representative backbones, such as LLaVA-NeXT or Qwen3-VL, to ensure the findings generalize to models with different architectural refinements.

4.The paper does not assess the sensitivity of Hyper-LLaVA to Task Order or Instruction Templates. In CIT, performance can vary significantly depending on the curriculum.

5.The proposed routing involves several operations: Task-aware Modality Weight Generation, Hyperbolic Task Similarity Estimation, and Dual-Modality Balanced Routing. The authors must provide a quantitative comparison of training time per task and inference time per sample against baselines to prove that the performance gains are not outweighed by excessive computational overhead.

---

> ### Author Rebuttal · Authors · 2026-03-31
>
> We sincerely appreciate your constructive feedback and recognition. Below are our responses point by point, which we hope will address your concerns effectively.
>
> **W1: Limitations**
>
> (1) Intra-task diversity: To demonstrate the impact of intra-task diversity, we compare our distribution-based intra-modality task prediction metric with the Euclidean distance-based strategy widely adopted in existing works. The results are shown in the following table:
>
> Intra-modality task predication metric|Aggregation strategy|MFN
> -|-|-
> Euclidean distance (Existing)|Average|56.75
> Distribution (Ours)|Average|62.67
>
> The results show that our method achieves a substantial improvement of **5.92%** compared with existing Euclidean distance-based approaches, verifying the effectiveness of modeling intra-task diversity, which provides richer distributional information for task prediction.
>
> (2) Inter-modality aggregation: Based on the above distribution-based intra-modality task prediction, we further compare our adaptive aggregation strategy with the commonly used average aggregation method. The results are presented below:
>
> Aggregation strategy|Intra-modality task predication metric|MFN
> -|-|-
> Average (Existing)|Distribution|62.67
> Adaptive (Ours)|Distribution|66.62
>
> The results demonstrate that our method achieves an additional improvement of **3.95%** over conventional average aggregation, validating the effectiveness of the proposed adaptive inter-modality aggregation mechanism, which alleviates the influence of ambiguous single-modality information.
>
> **W2: Parameter growth design**
>
> (1) Introducing a new set of LoRA modules while retaining previously learned LoRA modules is a **classical design** in the continual learning literature, which has been shown to achieve a favorable balance between computational overhead and performance. Therefore, in this work, we follow this widely adopted baseline setting to ensure a **fair comparison** with existing methods.
>
> (2) Designing sub-linear parameter growth mechanisms represents an **insightful and largely underexplored research direction** in continual instruction learning. In future work, we plan to investigate parameter compression strategies based on inter-task knowledge distributions, aiming to reduce storage overhead while maintaining model performance.
>
> **W3: Different backbone**
>
> We conduct additional experiments using the LLaVA-NeXT backbone and compare against HiDe-LLaVA and SEFE:
>
> Method|MFN|MAA
> -|-|-
> HiDe-LLaVA|63.55|78.41
> SEFE|67.55|79.82
> Ours|73.93|82.91
>
> These results show that our method outperforms state-of-the-art SEFE by **6.38%** and **3.09%** on MFN and MAA, respectively, demonstrating robustness across different backbone architectures.
>
> **W4: curriculum**
>
> (1) Task orders. We further evaluate performance under different task sequences in the UCIT benchmark (ImgNetR, Arxiv, QAVizWiz, IconQA, CLEVR, Flickr). The MFN results are shown below:
>
> Method|RAVICF|AIRFCV|IFRCAV
> -|-|-|-
> HiDe-LLaVA|64.19|63.56|64.87
> SEFE|66.54|66.64|65.51
> Ours|72.81|75.01|72.40
>
> Our method outperforms SEFE by **6.27%/8.37%/6.89%**, respectively, demonstrating strong robustness across different task orders.
>
> (2) Instruction templates. (i) The scenario in which tasks share the same visual sources while differing in instruction templates is already included in the CoIN benchmark (Tab. 1 of our paper). Specifically, images across its sub-datasets overlap but are paired with different instructions, introducing substantial instruction variation.
> (ii) The results in Tab. 1 show that, even under significant instruction variability, our method achieves improvements of **3.65%** and **5.75%** in MFN and MAA, respectively. These gains verify the effectiveness of our approach in adapting to diverse instruction templates.
>
> **W5: Overhead**
>
> Please refer to Reviewer wLBo-W1 for a detailed analysis of overhead.
>
> **Q1, Q3: Typo**
>
> Thanks sincerely for pointing these typos out. We have carefully corrected them in our revised version.
>
> **Q2: Highlight**
>
> We did not intentionally highlight MFT and BWT to avoid potential misinterpretation.
>
> (1) MFN and MAA reflect the overall effectiveness of the model throughout the continual training process and therefore better represent its **practical utility in real-world applications**.
>
> (2) MFT and BWT, in contrast, measure **specific and partially decoupled aspects** of continual learning behavior. From a theoretical perspective, a model that ignores historical tasks may achieve an upper bound in MFT, while a model that underfits new tasks can artificially inflate BWT. Hence, improvements in these metrics do not necessarily indicate better overall performance.
>
> (3) To ensure a fair and practically meaningful comparison, we emphasize the comprehensive metrics (MFN and MAA) while still reporting MFT and BWT for completeness and transparency. A clarification note has been added to the captions of the revised tables to explicitly explain this formatting choice.

---

> > ### Author Rebuttal · Reviewer_tw6i · 2026-04-03
> >
> > I thank the authors for their responses. All my concerns have been addressed, and I have no further questions.

---

> > > ### Author Response · Authors · 2026-04-04
> > >
> > > Dear Reviewer tw6i,
> > >
> > > We sincerely thank you for your thorough and constructive review. Your insightful comments have been instrumental in improving our paper, and we appreciate the time and effort you've dedicated to it.
> > >
> > > We are pleased to have carefully addressed the concerns raised in the initial review within our rebuttal. Given that all technical concerns have now been resolved, we would be grateful if you would consider raising your score.
> > >
> > > Thank you again for your thoughtful engagement.
> > >
> > > Best regards,
> > >
> > > The Authors

---

### Official Review · Reviewer_eCkw · 2026-03-12

**Soundness:** 3
**Presentation:** 3
**Significance:** 3
**Originality:** 3
**Overall Recommendation:** 4
**Confidence:** 3

**Summary:**

This paper proposes Hyper-LLaVA, a novel framework that enhances routing capacity through hyperbolic uncertainty-aware modeling by embedding samples and task distributions into hyperbolic space to better capture intra-modality similarity via Poincaré distance, while simultaneously introducing an adaptive modality-balancing mechanism that quantifies task ambiguity using Bhattacharyya Distance to dynamically weight visual and textual modalities during inference. By jointly optimizing intra-modality matching and inter-modality fusion, Hyper-LLaVA effectively mitigates the "gravity well" effect and modality noise, demonstrating superior performance over state-of-the-art methods across challenging MCIT benchmarks with significant improvements in accuracy and stability.

**Compliance With Llm Reviewing Policy:**

Affirmed.

**Final Justification:**

The rebuttal addressed my main concerns

**Key Questions For Authors:**

● The show cases merely display the final results for each query, it would benefit if the author show the actual output of the MLLM.

● More see weaknesses.

**Limitations:**

The method's reliance on strong Gaussian assumptions, the lack of computational cost analysis, and the somewhat heuristic nature of the hyperbolic mapping prevent it from being fully convincing in its current form. The authors need to validate their distributional assumptions, provide efficiency metrics, and offer a more rigorous justification for why hyperbolic space is superior to advanced Euclidean uncertainty metrics.

**Strengths And Weaknesses:**

Strengths:

● The paper provides a rigorous theoretical justification for using Hyperbolic space (Poincaré ball).

● The experiment section provides comprehensive ablation on modality imbalance. The experimental section does not merely report aggregate metrics, it deeply investigates the cause of failure in existing methods.

Weaknesses:

● For the assumptions in modality weighting, the performance gains attributed to the "adaptive balancing" may be fragile if the underlying distributional estimates are inaccurate. The paper provides no empirical verification (e.g., via normality tests or visualization of actual vs. fitted distributions) to justify this strong statistical assumption.

● Insufficient ablation on hyperparameter sensitivity, while Figure 4(a) shows a bell-shaped curve for γ, the analysis is limited to a single benchmark setting. Given that the adaptive weighting relies on these temperatures to align dynamic ranges, poor tuning here could negate the benefits of the BC-based weights. The robustness of the method across different task sequences or domain shifts without re-tuning γ remains unproven.

● In theorem A.1, the connection between task variance and the optimal placement near the boundary is heuristic. The mapping function in Eq. 8-9 arbitrarily links aggregate variance to the radius. There is no theoretical guarantee that minimizing Poincaré distance in this specific construction correlates better with downstream classification accuracy than other uncertainty-aware metrics in Euclidean space (e.g., Mahalanobis distance).

---

> ### Author Rebuttal · Authors · 2026-03-31
>
> We sincerely appreciate the reviewer's constructive feedback and recognition. We hope the following responses address your concerns effectively.
>
> **W1: Statistical assumption**
>
> In the following, we provide a detailed verification and analysis of the statistical assumptions adopted in this work.
>
> (1) Independent distributions across dimensions. (i) In https://anonymous.4open.science/r/Hyper-DDD/Fig-2.pdf, we visualize the covariance matrices of visual and instruction modalities across feature dimensions. The results show that covariance values are primarily concentrated along the diagonal, indicating that different feature dimensions are approximately mutually independent. (ii) The assumption of approximately independent feature dimensions is widely adopted in prior literature to achieve a practical balance between modeling performance and computational overhead [1,2]. Under this assumption, the computational complexity can be reduced from $O(n^2)$ to $O(n)$.
>
> (2) Approximately Gaussian distributions. (i) In https://anonymous.4open.science/r/Hyper-DDD/Fig-3.pdf, we present the statistical distributions of image and instruction features across different dimensions. The results indicate that features within each task approximately follow Gaussian distributions. (ii) The assumption that feature representations approximately follow Gaussian distributions is also commonly adopted in prior works [1,2], as real-world high-dimensional representations often exhibit near-Gaussian statistics after deep feature extraction.
>
> To summarize, the statistical assumptions adopted in this work are consistent with widely accepted practices in the literature and are further supported by empirical observations from real data statistics.
>
> [1] Probabilistic Embeddings for Cross-Modal Retrieval. CVPR2021
>
> [2] Distribution-aware knowledge prototyping for non-exemplar lifelong person
>  re-identification. CVPR2024
>
> **W2: Hyperparameter sensitivity**
>
> (1) In https://anonymous.4open.science/r/Hyper-DDD/Fig-4.pdf, we present the hyperparameter sensitivity analysis on the UCIT benchmark across different task sequences. The results show that, under varying experimental conditions, the optimal value of $\gamma$ remains consistent with the observations on the CoIN benchmark (Fig. 4 (a) of our paper), demonstrating the stability and generalizability of $\gamma$.
>
> (2) Please refer to Reviewer wLBo-W2 for the theoretical analysis explaining why $\gamma$ achieves optimal performance when $\gamma \approx 1$.
>
> **W3: Theoretical guarantee**
>
> We thank the reviewer for the insightful comment. This design is not arbitrary but geometrically principled, and we provide new evidence against Mahalanobis distance to support it.
>
> (1)Why Hyperbolic Mapping over Euclidean Uncertainty (Mahalanobis)?
> To directly answer your question, we conducted a new ablation experiment replacing our Poincaré routing with the Mahalanobis distance. The MFN dropped significantly to **58.32%** (compared to our **66.62%** and the naive Euclidean's 56.75%).
>
> (2) The Theoretical Flaw of Mahalanobis in MCIT.
> The failure of Mahalanobis distance in our setting exposes its critical limitation: it penalizes variance linearly in a flat space. For a test sample evaluated against a high-variance general task, the large covariance matrix in the denominator of the Mahalanobis distance artificially shrinks the distance, creating an even stronger "gravity well." Conversely, for zero-variance tasks, the inverse covariance becomes singular, leading to severe numerical instability.
>
> (3) The Geometric Guarantee of our Construction.
> Our mapping (Eq. 8-9) resolves this by leveraging the unique topology of the Poincaré ball. By mapping small variances to the boundary ($r \to 1$), we exploit the Gradient Explosion at the Boundary (Theorem A.2). The distance metric $(1 - \|z_k\|^2)$ in the denominator strictly enforces that a sample must be perfectly aligned to be assigned to a specific task, imposing an exponential penalty on mismatches rather than a linear one. Furthermore, the exponential volume growth guarantees that an arbitrary number of specific tasks can be packed near the boundary without overlapping.
>
> **Q1: Case visualization**
>
> (1) The displayed results in Fig. 5 and Fig. 9 of our paper are exactly the outputs generated by the MLLM. This is because the MLLM outputs in the MCIT benchmark are restricted to a single word or a predefined option specified in the input instructions.
>
> (2) Such a design in the MCIT benchmark is intentionally adopted to facilitate standardized and quantitative evaluation across different methods.
>
> **Q2: Efficiency metrics**
>
> Please refer to our response to Reviewer wLBo-W1, where a detailed analysis of the efficiency metrics, including runtime, GPU memory, storge memory and parameter overhead, is provided.

---

> > ### Author Rebuttal · Reviewer_eCkw · 2026-04-04
> >
> > Thanks for the careful response. All my concerns are addressed, and I will update my score

---

> > > ### Author Response · Authors · 2026-04-04
> > >
> > > Dear Reviewer eCkw,
> > >
> > > Thank you for your time and expertise in evaluating our work. We sincerely appreciate your positive assessment and recommendation. Your insightful suggestions have significantly improved the comprehensiveness and clarity of our manuscript.
> > >
> > > Thank you again for your thoughtful engagement.
> > >
> > > Best regards,
> > >
> > > The Authors

---

### Official Review · Reviewer_nyeV · 2026-03-12

**Soundness:** 3
**Presentation:** 4
**Significance:** 3
**Originality:** 4
**Overall Recommendation:** 4
**Confidence:** 3

**Summary:**

This paper studies multimodal continual instruction tuning (MCIT), with a particular focus on expert routing in LoRA-based mixture-of-experts style systems. The paper proposes Hyper-LLaVA, which represents task distributions in hyperbolic space for more robust intra-modality matching and uses Bhattacharyya-distance-based modality weighting to balance visual and textual routing scores adaptively.

**Compliance With Llm Reviewing Policy:**

Affirmed.

**Key Questions For Authors:**

1. The method clearly improves MFN and MAA, but BWT is still worse than ModalPrompt. So in the end, would you say the gain mainly comes from better learning of new tasks, or from actually reducing forgetting?

2. A lot of the weighting design seems to rely on the assumption that task features are roughly Gaussian with independent dimensions. How sensitive is the method if the real feature distributions are multi-modal or strongly non-Gaussian?

3. The current experiments use LLaVA-v1.5-7B and two benchmarks. Do you have any evidence that the routing mechanism still works as well with a different backbone, or under different task orders?

**Limitations:**

yes

**Strengths And Weaknesses:**

The paper has a clear motivation and a fairly coherent method design. The central problem is well framed: routing based only on task centers can miss intra-task diversity, and equal-weight cross-modal fusion can be brittle when one modality is less reliable.

The main weakness is that the paper’s novelty feels more like a strong combination of known ideas than a clearly new paradigm, since the overall framework still follows the familiar recipe of task-specific LoRA experts plus routing, with the main changes being hyperbolic matching and uncertainty-aware fusion. In addition, some key parts of the method rely on fairly strong modeling assumptions, especially the use of approximately Gaussian and independent feature distributions when computing Bhattacharyya distances and justifying the weighting rule.

---

> ### Author Rebuttal · Authors · 2026-03-31
>
> Thanks for your constructive feedback and recognition. We hope the following responses address your concerns.
>
> **W1: Novelty**
>
> (1) In the MCIT task, vanilla LoRA has shown promising knowledge acquisition capacity in early works, but the key challenge lies in **LoRA selection error during testing**.
>
> (2) Existing works usually introduce **additional constraints** on vanilla LoRA to mitigate the influence of LoRA selection errors. However, they suffer from both limited learning capacity and limited LoRA selection accuracy.
>
> (3) Our key novelty lies in improving LoRA selection accuracy from the **mathematical perspective** without sacrificing the learning capacity of vanilla LoRA. Specifically, (i) we pioneeringly provide a systematic investigation of the intra-modality and inter-modality **distribution characteristics** and underscore the overlook of intra-modality task-wise distribution boundaries and inter-modality ambiguity in existing works. (ii) To address these limitations, we propose a novel paradigm incorporating distribution-aware intra-modality task prediction with inter-modality adaptive fusion strategies, grounded in theoretical analysis. Experimental results show that this framework exhibits substantial improvement of over **3.11%** on the MFN and MAA metrics, verifying the contribution of this work.
>
> **W2, Q2: Modeling assumptions**
>
> (1) Independent distributions across dimensions. (i) The assumption of approximately independent feature dimensions is widely adopted in prior literature to achieve a practical balance between modeling performance and computational overhead [1,2]. Under this assumption, the computational complexity can be reduced from $O(n^2)$ to $O(n)$. (ii) In https://anonymous.4open.science/r/Hyper-DDD/Fig-2.pdf, we visualize the covariance matrices of visual and instruction modalities across feature dimensions. The observed covariance values are concentrated along the diagonal, indicating that different feature dimensions are approximately independent.
>
> (2) Approximately Gaussian distributions. (i) The assumption that feature representations approximately follow Gaussian distributions is also commonly adopted in prior works [1,2], as real-world high-dimensional representations often exhibit near-Gaussian statistics after deep feature extraction. (ii) In https://anonymous.4open.science/r/Hyper-DDD/Fig-3.pdf, we present the statistical distributions of image and instruction features across different dimensions. The results indicate that features within each task approximately follow Gaussian distributions.
>
> [1]Probabilistic Embeddings for Cross-Modal Retrieval. CVPR2021
>
> [2]Distribution-aware knowledge prototyping for non-exemplar lifelong person
>  re-identification. CVPR2024
>
> **Q1: Metric**
>
> (1) Compared with ModalPrompt, the performance gain mainly stems from improved learning of new tasks. This is primarily because ModalPrompt imposes strict constraints on model updates to mitigate forgetting. While this design improves BWT, it restricts model plasticity and consequently limits performance on key metrics (MFN and MAA), leading to over **9.0%** degradation compared with our method as shown in Tab. 1 and Tab. 2 of our paper.
>
> (2) Compared with state-of-the-art LoRA-based methods HiDe-LLaVA and SEFE, our method exhibits substantial improvements of over **3.9%** in BWT across different benchmarks. These gains arise because our method ensures correct LoRA selection during inference, thereby effectively alleviating performance degradation (forgetting) of historical tasks.
>
> **Q2: Feature distributions**
>
> (1) When the data distributions are non-Gaussian, our method can be naturally extended by modeling each task distribution as a Gaussian mixture rather than a single Gaussian. In this case, task prediction is obtained by aggregating the contributions of multiple Gaussian components, enabling flexible adaptation to non-Gaussian data distributions.
>
> (2) We also construct a subset with a non-Gaussian (Laplace) distribution sampled from the original MCIT benchmark. The results demonstrate that our method outperforms the state-of-the-art SEFE by **8.72%** and **5.77%** in MFN and MAA, verifying our robustness to non-Gaussian distribution.
> |Method|MFN|MAA
> |-|-|-
> |HiDe-LLaVA|51.41|52.33
> |SEFE|53.49|59.67
> |Ours|62.21|65.44
>
> **Q3: Additional experiments**
>
> (1) When adopting LLaVA-NeXT backbone suggested by Reviewer tw6i, our method outperforms state-of-the-art SEFE by **6.38%** and **3.09%** on MFN and MAA, verifying our robustness to backbones.
> |Method|MFN|MAA
> |-|-|-
> |HiDe-LLaVA|63.55|78.41
> |SEFE|67.55|79.82
> |Ours|73.93|82.91
>
> (2) We have conducted additional experiments for different orders of tasks in UCIT (ImgNetR, Arxiv, QAVizWiz, IconQA, CLEVR, Flickr). The results show that our method outperforms state-of-the-art SEFE by **6.27%/8.37%/6.89%** on MFN and MAA, respectively.
> Method|RAVICF|AIRFCV|IFRCAV
> |-|-|-|-
> |HiDe-LLaVA|64.19|63.56|64.87
> |SEFE|66.54|66.64|65.51
> |Ours|72.81|75.01|72.40

---

> > ### Author Rebuttal · Reviewer_nyeV · 2026-04-05
> >
> > My comments have been addressed. I will maintain my positive score.

---

> > > ### Author Response · Authors · 2026-04-05
> > >
> > > Dear Reviewer nyeV,
> > >
> > > We sincerely appreciate your thoughtful feedback and the time you dedicated to reviewing our work. Your insightful comments have been invaluable in refining our presentation and strengthening the manuscript. We are grateful for the opportunity to clarify our approach and truly appreciate your recognition of our work.
> > >
> > > Thank you again for your thoughtful engagement.
> > >
> > > Best regards,
> > >
> > > The Authors

---

### Official Review · Reviewer_wLBo · 2026-03-13

**Soundness:** 3
**Presentation:** 3
**Significance:** 3
**Originality:** 3
**Overall Recommendation:** 5
**Confidence:** 4

**Summary:**

This paper addresses the problem of Multimodal Continual Instruction Tuning (MCIT) for large multimodal models, focusing on two key limitations of existing methods: inadequate modeling of intra-task distributional diversity and unreliable inter-modality aggregation under task ambiguity. The authors propose Hyper-LLaVA, a framework that introduces hyperbolic task similarity estimation to capture intra-task uncertainty and a modality-balancing mechanism based on the Bhattacharyya distance to adaptively weight visual and instruction modalities during expert routing. Experiments on two MCIT benchmarks (CoIN and UCIT) demonstrate that Hyper-LLaVA outperforms state-of-the-art methods across multiple metrics (MAA, MFN, MFT, BWT), with particularly strong improvements.

**Compliance With Llm Reviewing Policy:**

Affirmed.

**Final Justification:**

I recommend that this paper be accepted by ICML.

**Key Questions For Authors:**

Although many existing large-model studies mainly rely on intuitive designs, this work is instead built upon solid theoretical foundations, which significantly strengthen its technical credibility and contribution. Overall, this is an impressive and well-executed study. The reviewer recommends that the authors further improve the paper by providing additional analyses on computational overhead, hyperparameter sensitivity, and training dynamics to enhance the completeness of the evaluation. The authors are recommended to provide a discussion on the computational overhead.

**Limitations:**

The authors are recommended to provide a discussion on the computational overhead.

**Strengths And Weaknesses:**

**Strengths**:
* The paper is well-structured and clearly written, with detailed theoretical derivations in the appendix and insightful visualizations.
* The paper identifies and addresses two fundamental and underexplored challenges in MCIT, intra-task diversity and inter-modality imbalance, with a well-motivated and technically sound approach.
* The use of hyperbolic geometry to model task uncertainty is novel and theoretically grounded, with the Poincaré ball providing a natural way to separate general and specific tasks via radius scaling based on variance.
* The modality-balancing mechanism based on Bhattacharyya distance is both intuitive and rigorous, with theoretical justification linking it to optimal inverse-variance fusion.
* Extensive experiments on two challenging benchmarks with comprehensive metrics and ablation studies provide strong empirical
support for the proposed method.

**Weaknesses**:
* Lack of overhead comparison. The computational overhead during both training and inference is not reported. A comparison of runtime, memory consumption, or parameter overhead with existing methods would provide a more complete evaluation of the practicality.
* Limited hyperparameter analysis. The hyperparameter γ i is empirically chosen but not systematically analyzed. A sensitivity analysis would help demonstrate the robustness of the method.
* Missing training dynamics analysis. The paper only reports the final performance after multiple training stages. It would be helpful to include the performance trajectory during training (e.g., stage-wise accuracy curves) compared with the baseline methods to illustrate how knowledge is accumulated throughout the continual learning process.

---

> ### Author Rebuttal · Authors · 2026-03-31
>
> Thank you for your constructive feedback and recognition. We hope the following responses address your concerns.
>
> **W1: Computational overhead**
>
> In the following table, we report the runtime (Training time per task, Inference time per sample), memory consumption (GPU memory-Train, GPU memory-Inference, per-task Storage memory of method-specific design), parameter overhead (Parameter) in comparison with state-of-the-art HiDe-LLaVA and SEFE (Due to time limitation, the experiments are conducted on the UCIT Benchmark):
>
> |Method|Training time per task (h)|Inference time per sample (s)|GPU memory-Train (GB)|GPU memory-Inference (GB)|Storge memory (GB)|Parameter (M)|
> |-|-|-|-|-|-|-|
> |LoRA (Baseline)|0.341|0.198|256|16|0.243|147.76
> |HiDe-LLaVA|0.345|0.211|256|16|0.243|147.76
> |SEFE|0.591|0.198|256|16|0.243|147.76
> |Ours|0.346|0.199|256|16|0.243|147.76
>
> Therefore, compared to the existing methods, our Hyper-LLaVA requires **comparable** overhead but achieves substantial MFN/MAA improvements of **3.65%/5.75%** according to Tab.2 of our paper.
>
> (1) The efficiency of training time per task is because the distribution estimation is achieved by calculating the mean and variance of training features, which could be achieved online during training without additionally modeling forward. Thus, it only brings 0.005 h additional training time, which is negligible compared to the LoRA baseline (0.341h).
>
> (2) As for the Inference time per sample, our method is the same as LoRA Baseline and more efficient than HiDe-LLaVA and comparable to SEFE. This is because the parameter selection process only involves simple computing and is negligible compared to the model forward. Besides, HiDe-LLaVA requires more inference time because they adopt multi-round prediction. Therefore, our method is both efficient and effective for inference.
>
> (3) Our method almost introduces no additional GPU memory for training and inference. This is because the distribution estimation and parameter selection only involve a few vectors and are extremely lightweight compared to the model forward.
>
> (3) Our method also almost introduces no  Storge memory/Parameter (M) for each task. This is because for each task, we only introduce four vectors to store the distribution information, which are negligible compared to the model.
>
> **W2: Hyperparameter analysis**
>
> The parameter $\gamma$ is introduced to scale the range of each task on the Poincaré ball after normalization. In the following, we provide a theoretical justification for why $\gamma \approx 1$ is an appropriate choice.
>
> Theorem 1. Equation (9) in our paper can be simplified as $r_k = e^{-\gamma x}$, where $x = \frac{V_k}{\overline{V}_t + \epsilon}$. When $\gamma \approx 1$, Eq. (9) exhibits optimal representation capacity.
>
> Proof. Treating $x$ as the variable, the derivative of $r_k$ is $g(\gamma,x)=-\gamma e^{-\gamma x}$. Further taking $\gamma$ as the variable, the derivative of $g(\gamma, x)$ becomes $g'(\gamma,x)=-e^{-\gamma x}+\gamma xe^{-\gamma x}$. Setting $g'(\gamma, x)=0$ yields $\gamma = 1/x$, at which point $g(\gamma, x)$ reaches its extremum, indicating the most stable sensitivity of $r_k$ with respect to variations in $x$.
>
> Since $\overline{V}_t$ denotes the moving average of $V_k$, the normalized variabl $x = \frac{V_k}{\overline{V}_t + \epsilon}$ is typically close to 1 during training. Consequently, the optimal condition $\gamma = 1/x$ implies $\gamma \approx 1$. Under this setting, Eq. (9) operates near its maximal representation capacity.
>
> This theoretical observation is consistent with the empirical results shown in Fig. 4 (a) of our paper.
>
> **W3: Training dynamics analysis**
>
> We have visualized the stage-wise accuracy curves in https://anonymous.4open.science/r/Hyper-DDD/Fig-1.pdf. The results show that our method achieves similar performance with state-of-the-art methods at the initial task but shows increasing advantages as the task increases. These results show that our approach exhibits significantly superior knowledge accumulation capacity throughout the continual learning process. These improvements arise from our uncertainty-aware modality-balanced routing design, which effectively improves the learned knowledge utilization capacity.

---

> > ### Author Rebuttal · Reviewer_wLBo · 2026-03-31
> >
> > The authors' rebuttal has effectively addressed my concerns and convinced me of the paper's strengths. Therefore, I stand by my assessment that this paper is highly deserving of acceptance at ICML, and I maintain my score.

---

> > > ### Author Response · Authors · 2026-04-01
> > >
> > > Dear Reviewer wLBo,
> > >
> > > Thank you for your time and effort in reviewing our paper. We sincerely appreciate your recognition of our novelty, theoretical contributions, clarity, and effectiveness, as well as your positive recommendation. Your insightful suggestions have significantly helped us improve the comprehensiveness of the experimental evaluation and strengthen the quality of the manuscript.
> > >
> > > Sincerely,
> > >
> > > The Authors

---

### Decision · Program_Chairs · 2026-04-30

**Decision:**

Accept (regular)

**Comment:**

The paper proposes to learn adaptive task similarity to capture intra-task distributional diversity as well as adaptive modality weighting. This addresses primary limitations of current mixture-of-experts in multimodal learning where modalities maybe equally weighted and task-similarity measures are insufficient. Studies on major benchmark so reasonable performance gain.

Reviewers all suggest that paper is well-written and addressing an important gap. Use of hyberbolic/adaptive poincare distance is novel and theoretically well motivated experiments are comprehensive and also highlight failure modes.

Reviewers had clarification questions regarding:
1. Hyperparameter tuning
2. Clarification/ablation of where performance gain is coming from
3. Insufficient consideration of task order and re-use of LoRA modules across tasks.

Authors satisfactorily addressed all reviewer concerns. As a result I finally recommend an Accept.